# Factored DRO: Factored Distributionally Robust Policies for Contextual Bandits

**Tong Mu**
Stanford university
`tongm@cs.stanford.edu`

**Yash Chandak**
University of Massachusetts
`ychandak@cs.umass.edu`

**Tatsunori Hashimoto**
Stanford University
`thashim@stanford.edu`

**Emma Brunskill**
Stanford University
`ebrun@cs.stanford.edu`

## Abstract

While there has been extensive work on learning from offline data for contextual multi-armed bandit settings, existing methods typically assume there is no environment shift: that the learned policy will operate in the same environmental process as that of data collection. However, this assumption may limit the use of these methods for many practical situations where there may be distribution shifts. In this work we propose Factored Distributionally Robust Optimization (Factored-DRO)[1], which is able to separately handle distribution shifts in the context distribution and shifts in the reward generating process. Prior work that either ignores potential shifts in the context, or considers them jointly, can lead to performance that is too conservative, especially under certain forms of reward feedback. Our Factored-DRO objective mitigates this by considering the shifts separately, and our proposed estimators are consistent and converge asymptotically. We also introduce a practical algorithm and demonstrate promising empirical results in environments based on real-world datasets, such as voting outcomes and scene classification.

## 1 Introduction

In many real-world applications we often have access to historical data about past decisions and their associated outcomes. This data can be leveraged for *off-policy learning* to optimize for a decision making rule, called a policy, that is more effective for the given application than that used for data collection. Off-policy learning from historical data for contextual multi-armed bandit settings has been well studied, due in part to the myriad of potential applications; from personalized recommendation [Strehl et al., 2010] to optimizing medical treatment [Kim et al., 2011]. Prior methods often focus on the stationary setting, where the learned policy will be deployed in a setting with the same underlying data generation process that produces the observed contexts, and rewards upon intervention [Precup, 2000, Liu et al., 2018, Kallus and Uehara, 2020a,b, Liu et al., 2020]. However, in many settings, this assumption is not true and the distributions may shift.

As a motivating example, consider the problem of increasing political participation during elections using various interventions to encourage voting. In the past, different strategies have been implemented, ranging from reminding people of their own voting history, to reminding them of the voting history of their neighborhood, etc. [Gerber et al., 2008]. Leveraging this historical data, an important question is: *what strategy should be used during the next election to maximize political participation*? Addressing this question requires careful consideration of two potential sources of

---

[1]Code: `https://github.com/StanfordAI4HI/FactoredDRO`

36th Conference on Neural Information Processing Systems (NeurIPS 2022).

distribution shift: **(a)** How much has the population demographic in the area changed since the past data was collected? (e.g. the percentage of single person households may have changed due to change and developments in the city.) **(b)** How much has the effectiveness of each strategy on each demographic group changed? (e.g. as technology develops, the effectiveness of mail-based interventions may change.) Such challenges are ubiquitous in real-world problems. For example, in automated healthcare, data from patients of one locality may be used to develop treatments for other localities where the effectiveness of treatments may differ [Finlayson et al., 2021]. Similarly, recommendation systems need to account for both the changing interests and the demographic shifts of their user base [Koh et al., 2021]. Developing a general procedure for addressing these challenges can provide a reliable solution for such applications where there may be high stakes in societal behavior, public health, or monetary assets.

Without considering potential shifts, there is a risk of using suboptimal strategies, however it is often difficult to predict exactly the shifts that will occur. Distributionally robust optimization formulations such as ours attempt to address this uncertainty over potential shifts by learning policies that attain good performance on all potential distributions within a user-defined set of distributions (denoted the *uncertainty set*). Prior methods for contextual multi-armed bandits [Si et al., 2020, Mo et al., 2021, Kallus et al., 2022] that aim to provide robustness against distribution shifts do not separate the different sources of shifts, and consider them in a joint formulation. These methods may be disadvantageous as they lack the flexibility to separate the sources. Consequently, we empirically find and show that some of these methods may produce degenerate results under binary reward feedback. This issue is theoretically proven to occur in the DRO literature in supervised learning [Hu et al., 2018]. Due to this, methods that better use structure to mitigate this issue have been an active area of research in the supervised learning DRO literature [Sagawa et al., 2019, Duchi et al., 2019]. Inspired by this, our insight is that better usage of structure is important, and there is often expert opinion about the maximum amount of distribution shift that may occur that can be used to achieve better performance on the deployment environments.

In this work we incorporate this insight in the framework of contextual bandits. We propose our Factored Distributionally Robust Optimization (FDRO) algorithm that uses past data to search for a policy that has the best worst-case performance, when the *context* (e.g., population demographic) or/and the *reward model* (e.g., influence of strategies on voting behavior) can be different from the ones during data collection. Towards this goal, we make three primary contributions:

I. We present a factored distributionally robust optimization formulation for contextual bandits that provides end-users the flexibility to *separately* specify different amounts of potential shifts in the two main sources of uncertainty: the context distribution and the reward distribution. Our factored formulation can overcome some failure cases of non-factored prior methods Si et al. [2020].

II. We establish theoretically that the proposed estimators provide convergence at a rate of $O\left(\sqrt{\log(n)/n}\right)$ in evaluation of any policy's worst-case performance under distribution shifts, as well as to a policy with the best worst-case performance.

III. Building on our theoretical insights, we develop a practical algorithm, FDRO, that uses function approximation and a Taylor series expansion. We show that FDRO provides promising empirical results on several domains based on real-world datasets.

## 2 Related Works

Both off-policy learning for bandits and reinforcement learning, as well as distributionally robust optimization (DRO) for supervised learning and stochastic optimization have rich histories of research. To briefly summarize, methods for off-policy evaluation/learning predominantly assume that the deployment environment for the learned policy will be the same as the environment the historical data was collected in [Precup, 2000, Jiang and Li, 2016, Thomas and Brunskill, 2016a,b, Gelada and Bellemare, 2019, Liu et al., 2018, Imani et al., 2018, Yang et al., 2020, Jiang and Huang, 2020, Kallus and Uehara, 2020a,b, Liu et al., 2020]. In contrast, DRO methods for tackling distributional shifts have predominantly focused on settings like supervised learning, and do not consider decision making aspects present in (contextual) bandits [Bertsimas and Sim, 2004, Hu and Hong, 2013, Namkoong and Duchi, 2016, Shapiro, 2017, Duchi et al., 2016, Duchi and Namkoong, 2018, Hashimoto et al., 2018, Duchi et al., 2019, Sagawa et al., 2019, Oren et al., 2019, Michel et al., 2022].

In addition to the prior work already previously mentioned in Section 1, there have been a few recent papers investigating DRO for use in bandits and reinforcement learning. Some works use DRO to account for the mismatch between the true distributions (of the context and rewards) and the observed empirical distributions [Swaminathan and Joachims, 2015a,b, Faury et al., 2020, Sakhi et al., 2020]. These methods account for the potential errors of finite-data estimation, and, under infinite data, they converge to the best training distribution policy. However we are concerned with the setting when the test distribution may be different, and the best training-distribution policy may not be optimal.

There has also been other work that consider settings when different forms of information are available a priori. For example, in settings where knowledge about the distribution of covariates during evaluation [Kato et al., 2020], bounds on the ratio of train-eval covariate distribution [Hatt et al., 2021, Giguere et al., 2021], bounds on train-evaluation performance differences [Chandak et al., 2021], the causal directed acyclic graph [Subbaswamy et al., 2019, Singh et al., 2021], or when there is data through time that can be used to model potential non-stationarity [Thomas et al., 2017, Chandak et al., 2020, Xie et al., 2020, Qin and Russo, 2021] have been considered. While relevant, they look at complementary problems.

## 3 Preliminaries

In this work, we consider the batch setting where the goal is to learn from an existing dataset $\mathcal{D}$. We consider a setting with discrete actions, where $\mathcal{A}$ denotes the set of $k$ discrete actions, $A$. Let $\mathcal{X} \subset \mathbb{R}^d$ denote the set of contexts, $X$. A policy $\pi$ denotes a mapping from a context $X$ to a distribution over actions. For a given context $X$ and action $A$ we observe a stochastic reward $R$. We assume the dataset, $\mathcal{D}$, consists of $n$ independently collected datapoints, each a (context, action, reward) triplet: $\mathcal{D} = \{X_i, A_i, R_i\}_{i=1}^n$. For this dataset, we assume the actions are selected with some data collection policy $\pi_0$ where $A_i \sim \pi_0(\cdot|X_i)$. We assume there is an underlying context distribution $P_{x_0}$, and for each context-action pair there is an underlying reward distribution $P_{r_0|X,A}$. We additionally assume the observed contexts, $X_i$ are independent and identically distributed (i.i.d.) draws from $P_{x_0}$, and the observed rewards are i.i.d. draws from $P_{r_0|X_i,A_i}$. We assume $P_{x_0}$ is known while $P_{r_0|X_i,A_i}$ is unknown.

We consider using $\mathcal{D}$ to learn a policy $\pi$ that will achieve the highest worst-case-reward when the deployment environment may be shifted from the one of data collection. Given a fixed policy, there are two ways the environment can shift and change the distribution of future outcomes. First the context distribution can shift so the inputs we see are different. Second the reward generation model can shift, changing the reward distribution of each (context, action) pair. We assume the exact shift is not known, so we would like to learn a policy that would do well on all environments within a user specified set of distributions of the context distribution $P_x$ and reward distributions $P_{R|X,A}$. Formally, given $\mathcal{D}$ we aim to find the policy that optimizes the following distributionally robust objective:

$$\pi^*_{FDRO} \in \underset{\pi \in \Pi}{\operatorname{argmax}} \inf_{P_x \in \mathbb{P}_{x_0}} \mathbb{E}_{P_x}\left[ \mathbb{E}_\pi \left[ \inf_{P_r \in \mathbb{P}_{r_0|X,A}} \mathbb{E}_{P_r}[R] \right] \right] \tag{1}$$

In Equation (1), the uncertainty sets $\mathbb{P}_{r_0|X,A}$ and $\mathbb{P}_{x_0}$ describe the set of possible distributions, $P_r$ and $P_x$, respectively. The optimization set $\mathbb{P}_{r_0|X,A}$ is conditioned on random variables $X \sim P_x$ and $A \sim \pi(\cdot|X)$. We term our objective *Factored-DRO* as we have two layers of distributionally robust optimization, first over reward shifts $P_{R|X,A}$, and another over context shifts $P_x$.

For theoretical analysis, we make the following assumptions common in prior literature [Si et al., 2020, Swaminathan and Joachims, 2015a, Kallus and Uehara, 2020a].

**Assumption 1.** *Assumptions on $\pi_0$ and the reward distribution*

*(i) Unconfounded: Rewards are independent with A conditioned on X: $(R_{X,A_1}, R_{X,A_2}...) \perp\!\!\!\perp A|X$*

*(ii) Bounded reward support: $0 \leq R \leq R_M$ for all rewards for a finite $R_M$*

*(iii) Coverage/Sufficient exploration: $\pi_0(A|X) \geq \epsilon_\pi$ for all $(X, A) \in \mathcal{X} \times \mathcal{A}$ for some $\epsilon_\pi > 0$*

*(iv) Positive density of rewards: Let $f_{R|X,A}$ be the pdf (continuous case) or $p_{R|X,A}$ be the pmf (discrete case) of the reward distribution $P_{R|X,A}$. Then assume, for all reward distributions involved, it holds that $f_{R|X,A} \geq \epsilon_r$ (continous) or $p_{R|X,A} \geq \epsilon_r$ (discrete) on the support of of R for all $(X, A) \in \mathcal{X} \times \mathcal{A}$ for some $\epsilon_r > 0$.*

We make an additional assumption, which we do not require in our practical algorithm:

**Assumption 2.** *Discrete contexts with sufficient mass: The context set $\mathcal{X}$ is a finite discrete set and the probability mass of each context is lower bounded with $P_{x_0}(x = X) \geq \epsilon_X$ for some $\epsilon_X > 0$*

**Note about model misspecification:** Context shifts are an issue when we have model misspecification and cannot model the best action for each context. In these cases, the learned model encodes the biases of the population distribution. Model misspecification can occur when the model is under-specified, for example, to reduce variance (bias-variance tradeoff), for interpretability, or due to unobserved variables. Misspecification also occurs when the model is over-specified, where it may overfit and not generalize (ex. neural networks [Sagawa et al., 2019]). The real world is often complex and difficult to model correctly, so throughout this work we assume the misspecified setting.

**Prior Work with Joint Shifts:** Recently Si et al. [2020] proposed an algorithm for solving a DRO formulation for contextual bandits that does not separate the context and reward shift. They consider the optimization problem: $\pi^* = \mathrm{argmax}_\pi \inf_{P \in \mathbb{P}_{P_0}} \mathbb{E}_P[R]$, where $P$ encapsulates the joint distribution of contexts and rewards and $\mathbb{P}_{P_0} = \{P : D_{kl}(P||P_0) \leq \delta\}$.

**Degenerate Outcomes with binary rewards under the joint objective:** Unfortunately, the joint formulation may have a key limitation in the common setting of binary reward outcomes (ex, clicked or not-clicked[Chu et al., 2011], voted or did-not-voted[Gerber et al., 2008], etc). In supervised learning (SL), similar DRO formulations have been theoretically proven to fail with certain loss functions. Hu et al. [2018] prove that when features are continous, and 0-1 classification loss is used, the DRO objective is equivalent to the standard, non-robust optimization of the loss function on the training distribution. Intuitively, DRO can be seen as changing the weights of the datapoints based on the incurred loss, and increasing the radius parameters increases the weights on datapoints with larger incurred losses. However, under 0-1 loss, DRO upweights all missclassified examples equally, which does not change the decision boundary. Consequently, in SL, methods that can mitigate this issue by using structure, such as Duchi et al. [2019] or Sagawa et al. [2019], have been proposed. We observe this issue empirically for the joint formulation in the binary reward case (see Section 6).[2] Intuitively, the lower reward points for each context action pair are all upweighted equally so the learned policy is equivalent to that optimized for the training distribution. Our method, which trades-off adding an additional hyperparameter with better structure usage to separate out the shifts mitigates this issue.

## 4 Factored DRO for Policy Evaluation and Learning

In our work, we consider uncertainty sets defined on the KL radius from the data collection distributions. Due to the joint decomposition, we require the user to specify two KL-radius hyperparameters, $\delta_c$ and $\delta_x$, for rewards and contexts respectively as opposed to a single hyperparameter required in prior work. The uncertainty sets defined in Equation 1 are defined as $\mathbb{P}_{r_0|X,A} = \{P_r : D_{kl}(P_r, P_{r_0|X,A}) < \delta_c\}$ and $\mathbb{P}_{x_0} = \{P_x : D_{kl}(P_x, P_{x_0}) < \delta_x\}$.

We can decompose our DRO objective from Equation 1 into two steps, first a policy evaluation step which given a policy evaluates the worst case policy reward. Notice the policy evaluation step itself can be separated into two steps: a step considering the potential reward generation shift ($Q_{X,A}$) and a step over the potential context shift to arrive at the desired value ($V(\pi)$):

$$V(\pi) = \inf_{P_x \in \mathbb{P}_{x_0}} \mathbb{E}_{P_x} \left[ \mathbb{E}_\pi \left[ Q_{X,A} \right] \right] \quad \text{where} \quad Q_{X,A} = \inf_{P_r \in \mathbb{P}_{r_0|X,A}} \mathbb{E}_{R \sim P_r}[R] \tag{2}$$

Second, we have a policy optimization step over the worst case policy rewards:

$$\pi^*_{FDRO} = \mathrm{argmax}_\pi V(\pi) \tag{3}$$

In this section, we present our estimators for policy evaluation and learning and our convergence guarantees. We give all result from prior work used and all full proofs in the appendix.

### 4.1 Policy Evaluation:

We start by discussing the policy evaluation step. Equation 2 as written is a difficult constrained optimization over a probability distribution space. Fortunately, prior work (such as Hu and Hong

---

[2]Note, Si et al. [2020] show good results in an environment (voting) which originally had binary rewards, but they transform the environment rewards to non-binary by shifting all rewards by a different offset per action.

[2013]) exists that uses methods from convex optimization to transform the problem to an equivalent but simpler convex optimization over a single scalar. They use convexity of the original objective and strong duality to derive an equivalent dual objective. We arrive at the following equivalent problem by applying this duality procedure twice, once for each layer.

**Theorem 1** (Strong Duality). *The optimization problem in equation 2 is equivalent to solving:*

$$V(\pi) = -\min_{\alpha_x > 0} \left\{ \alpha_x \log \mathbb{E}_{X \sim P_{x_0}} \left[ \exp \left( \frac{\mathbb{E}_\pi [-Q_{X,A}]}{\alpha_x} \right) + \alpha_x \delta_x \right] \right\}, \tag{4}$$

$$where \quad Q_{X,A} = -\min_{\alpha_c > 0} \left\{ \alpha_c \log \mathbb{E}_{R \sim P_{r_0 | X, A}} \left[ \exp \left( \frac{-R}{\alpha_c} \right) \right] + \alpha_c \delta_c \right\}. \tag{5}$$

*Proof Sketch* This follows from applying Theorem 1 of Hu and Hong [2013] twice.

We will use this easier to optimize dual form to derive estimators for policy evaluation. We will first consider the potential shift of the reward distribution of each (context, action) pair and calculate estimates of $Q_{X,A}$ of Equation 5. We will then use these values to perform the second DRO layer over context shifts to estimate $V(\pi)$ (Equation 4).

### 4.1.1 First DRO, Reward Shift:

For theoretical analysis, we consider the setting when the context and action sets are both finite and discrete (Assumptions 1.2 and 2). Therefore we have a finite number of unique $(X, A)$ pairs, each which occurs with a non-zero probability. For a given $(X, A)$ pair, let $\mathcal{R}_{X,A}$ correspond to the set of the observed reward $R_i$ every time $(X, A)$ is observed in $\mathcal{D}$: $\mathcal{R}_{X,A} = \{ R_i : (X_i, A_i, R_i) \in \mathcal{D}$ if $X_i = X, A_i = A\}$. Additionally, let $m_{xa} = p_0(x, a)n$ denote the expected number of datapoints in each $(X, A)$ bin and recall that $\delta_c$ denotes the user specified KL-radius of the ambiguity set.[3] For every unique $(X, A)$ we propose the following estimator for worst case context-action reward for $Q_{X,A}$ in equation 5:

$$\hat{Q}_{X,A} = -\min_{\alpha_c > 0} \left\{ \alpha_c \log \left( \frac{1}{m_{xa}} \sum_{R_j \in \mathcal{R}_{X,A}} \left[ \exp \left( \frac{-R_j}{\alpha_c} \right) \right] \right) + \alpha_c \delta_c \right\} \tag{6}$$

Note that the objective inside the minimization in equation 6 is convex in $\alpha_c$ for $\alpha_c > 0$, therefore many optimization methods (ex. gradient descent, bisection search) are able to achieve the global optimum when $\alpha_c \neq 0$. We give the gradients and a proof of convexity in the appendix. Additionally, note the $\hat{Q}_{X,A}$'s only depend on $\mathcal{D}$ and do not depend on $\pi$, therefore they only need to be computed once and can be used throughout the rest of the policy evaluation and optimization procedure.

### 4.1.2 Second DRO, Context Shift:

With the $\hat{Q}_{X,A}$ values estimated above for each $(X, A)$ pair we can complete the policy evaluation. Let $\delta_x$ denote the user-specified KL-radius for context distribution shift, then with the computed $\hat{Q}_{X,A}$ for each $(X, A)$ pair, we can complete the policy evaluation and estimate $V(\pi)$ from equation 2.

$$\hat{V}(\pi) = -\min_{\alpha_x > 0} \left\{ \alpha_x \log \sum_{X \in \mathcal{X}} P_{x_0}(x) \exp \left( \frac{-\sum_{a \in \mathcal{A}} \pi(a|X_i) \hat{Q}_{X,a}}{\alpha_x} \right) + \alpha_x \delta_x \right\} \tag{7}$$

We can show that when $\delta_c > 0$ and $\alpha_c^* > 0$ policy evaluation converges at rate $O(\sqrt{\log(n)/n})$:

**Theorem 2** (Convergence of policy evaluation). *For $n \geq \frac{(2 \log(2|\mathcal{X}||\mathcal{A}|/\delta)}{(\underline{p}_0(x,a))^2}$, the following holds for any $\pi$ with probability of at least $1 - \delta$:*

$$\left| V(\pi) - \hat{V}(\pi) \right| \leq \mathcal{O} \left( c \sqrt{\frac{\log(n) \log \left( \frac{2|\mathcal{X}||\mathcal{A}|}{\delta} \right)}{n}} \right) \quad where \quad c = \frac{32(R_{max})^3}{\delta_c^2 \underline{\alpha}_c^2 \exp(\frac{-R_{max}}{\underline{\alpha}_c}) \underline{p}_0(x,a)} \tag{8}$$

---

[3]Note it is also possible to provide a different $\delta_{c,(X,A)}$ for every unique $(X, A)$, or a different $\delta_{c,A}$ per action, however this greatly increases the hyperparameter burden on the end-user.

and $\underline{p}_0(x,a) = \min_{(x,a)} p_0(x,a) \geq \epsilon_\pi \epsilon_X$ *is the minimum over probability of occurrence of (X,A) pairs, and $\underline{\alpha}_c$ is the minimum value for $\alpha_c$.*

*Proof sketch:* We utilize a uniform convergence result from stochastic optimization and local Lipschitz-continuity to prove the desired result.

Note the objective in equation 7 is also convex and easy to optimize for $\alpha_x > 0$.

## 4.2 Policy Learning

We now consider the optimization over policy of equation 3. Define the true robust optimal policy as $\pi^* = \pi^*_{FDRO} = \arg\max_\pi V(\pi)$, with true optimal robust value $V^* = V(\pi^*)$. Define the empirical robust optimal policy as $\hat{\pi}^* = \arg\max_\pi \hat{V}(\pi)$ with empirical robust optimal value $\hat{V}^* = V(\hat{\pi}^*)$. We additionally define the set of $\epsilon$-optimal policies $\Pi^*_\epsilon = \{\pi : V(\pi) \geq V^* - \epsilon\}$. We then have the following statement of the convergence of optimal policy and value:

**Theorem 3** (Convergence of Policy Learning). *For $n \geq \frac{(2\log(2|\mathcal{X}||\mathcal{A}|/\delta)}{(\underline{p}_0(x,a))^2}$, the following holds with probability of at least $1 - \delta$:*

$$\left| \hat{V}^* - V^* \right| \leq \mathcal{O}\left( c\sqrt{\frac{\log(n)\log\left(\frac{2|\mathcal{X}||\mathcal{A}|}{\delta}\right)}{n}} \right) \tag{9}$$

*Equivalently this says*

$$P(\hat{\pi} \in \Pi^*_\epsilon) > 1 - \mathcal{O}\left( 2|\mathcal{X}||\mathcal{A}| \exp\left( -\frac{n\epsilon^2}{\log(n)c^2} \right) \right) \tag{10}$$

While this policy optimization is generally non-convex, for many policy classes we can use gradient ascent with multiple restarts. To learn a policy, we first compute the $\hat{Q}_{X,A}$ values of Section 4.1.1, we then perform alternating steps of the evaluation step in Section 4.1.2 and the policy optimization in this section (Section 4.2).

## 5 Practical Algorithm with Function Approximation

Our estimator for solving for the worst case reward shift value, $\hat{Q}_{X,A}$ for each context action pair in our first DRO step (section 4.1.1) relies on having a sufficient number of datapoints per each $(X, A)$ to approximate $\mathbb{E}_{R \sim P_{R|X,A}}[\exp(-R/\alpha_c)] = 1/m_{xa} \sum_j \exp(-R_j/\alpha_c)$. However, in many settings, contexts may not be discrete and, in these cases, each continuous context may only be observed once. One approach can be to discretize the context space, however selecting the correct discretization is challenging. Too coarse a discretization introduces looseness and leads to overly conservative results, but too fine a discretization may not leave enough items in each bin to accurately estimate $\mathbb{E}_{R \sim P_{R|X,A}}[\exp(-R/\alpha_c)]$. To handle continuous and/or large contexts spaces, instead of binning we use a function approximation that may not have the same convergence guarantees. Fortunately, as we show in Section 6 it can be quite accurate and lead to better results than the baseline methods. We develop an empirically stable method for estimating $\mathbb{E}_{R \sim P_{R|X,A}}[\exp(-R/\alpha_c)]$ using a Taylor series approximation. The key feature of our method is that it does not require fitting the exponential dependence on $R$ and $1/\alpha_c$. We found alternate methods, such as fitting an approximator for $\mathbb{E}_{R \sim P_{R|X,A}}[\exp(-R/\alpha_c)]$ directly by fitting $\exp(-R_i/\alpha_c)$ using corresponding values of $X_i$, $A_i$ and multiple values of $\alpha_c$ value sampled from the relevant range, to be unstable at small values of $\alpha_c$ due to the exponential dependence.

### 5.1 Approximating $\mathbb{E}_{R \sim P_{R|X,A}}[\exp(-R/\alpha_c)]$:

We can use the Taylor series expansion, $\sum_{i=0}^\infty \frac{(1/\alpha_c)^i}{i!} \mathbb{E}[(-R)^i]$, to estimate an order $m$ approximation. We do this by fitting a multi-output function approximator, $\mathcal{F}$ to approximate the first $m$ moments. We train $\mathcal{F}$ such that the $j^{th}$ output predicts $(-R_i)^j$ from $(X_i, A_i)$. When trained on all

|  | Simulated | Voting | Scene |
|---|---|---|---|
| $n_{total}$, $(n_{train}, n_{test})$ | $40K$, $(20K, 20K)$ | $180K$, $(\sim 44K, \sim 12K)$ | $2.407K$, $(\sim 1.6K, \sim 0.8K)$ |
| $|\mathcal{A}|$, $n_{cgs}$, $n_{feats\ used}$ | $4, 4, 5$ | $5, -, 2$ | $6, 2, 294$ (projected to 3) |
| Train % per cg | $(0.1, 0.4, 0.1, 0.4)$ | - | $(0.7, 0.3)$ |
| Test % per cg | Various | - | $(0.15, 0.85)$ |
| $\pi_0(a)$ | $(.1, .2, .4, .3)$ | $(.56, .11, .11, .11, .11)$ | $(.17, .17, .17, .17, .17, .17)$ |
| $\mathbb{E}_{train}[R|cg, A]$ | $cg_{1,3}:(.35, .5, .2, .14)$ | - | $cg_0:(.25, 0, 0, 0, .36, .34)$ |
|  | $cg_{2,4}:(.35, .2, .5, .14)$ | - | $cg_1:(0, .3, .36, .35, .1, 0)$ |

Table 1: Environment/Experiment details. The first row lists the approximate number of points in the train and test splits as we randomize the exact splits across runs. 'cg' stands for *coarse group* as described in the text. The last three rows list the percentages/reward values per action in the tuple. Note for voting, we do not have coarse groups as we use a shift between cities, additionally we consider using data from only some of the cities so not all data is used. We use random projection to project the feature space of Scene to avoid ill-conditioned behavior.

the data, $\mathcal{F}[j]$ should model the $j^{th}$ moment $\mathbb{E}_{P_{R|X_i,A}}[(-R)^j]$. Let $\mathcal{D}_X$ represent the sequence of contexts in the dataset. Then for every context $X_i \in \mathcal{D}_X$ and every $a \in \mathcal{A}$ and any $\alpha_c$, we use our approximators $\mathcal{F}_j$ to estimate the desired value:

$$\hat{\mathbb{E}}_{P_{R|X_i,a}}\left[\exp\left(\frac{-R}{\alpha_c}\right)\right] = 1 + \frac{1}{\alpha_c}\mathcal{F}_1(X_i, a) + .. + \frac{1}{\alpha_c^m}\frac{\mathcal{F}_m(X_i, a)}{m!}, \quad \text{for any } a \in \mathcal{A}, \alpha_c > 0. \quad (11)$$

This approximation achieves better regularity as we are no longer fitting the dependence on $\alpha_c$. Practically, we find order $m = 5$ can often be quite accurate. Additionally, note that while the context is high dimenstional, the Taylor expansion is only for the scalar $\alpha_c$.

**Complete Algorithm:** With our estimator $\hat{\mathbb{E}}_{P_{R|X_i,A}}[\exp(-R/\alpha_c)]$ we proceed with our method by first calculating $n \times k$ values of $\hat{Q}_{X_i,a}$ for each context in the dataset and each action in the action set. To do this, we iterate through $(X_i, a) \in (\mathcal{D}_X \times \mathcal{A})$ and solve the following *convex* problem in $\alpha_c$:

$$\hat{Q}_{X_i,a} = -\min_{\alpha_c > 0}\{\alpha_c \log(\hat{\mathbb{E}}_{P_{R|X_i,a}}[\exp(-R/\alpha_c)]) + \alpha_c \delta_c\}. \quad (12)$$

We then carry out the procedure detailed in Section 4.1.2 onwards with these $\hat{Q}_{X_i,a}$. Algorithm 1 in the Appendix provides our full algorithm for policy evaluation and optimization, that considers both the discrete binned case (Section 4.2) and the function approximation case just discussed.

# 6 Experiments

We test and analyze our algorithm empirically in three settings, including two derived from real data. For all experiments, we use the practical algorithm discussed in Section 5. While our method works for any policy that can be optimized, we consider learning the parameters $\theta$ of a stochastic softmax linear contextual bandit policy of the form:

$$\pi_\theta(X, A) = \frac{\exp(\phi(X, A)^T \theta)}{\sum_{a \in \mathcal{A}} \exp(\phi(X, a)^T \theta)} \quad (13)$$

This policy $\pi_\theta$ is differentiable so we optimize $\theta$ using gradient descent. For the function approximators $\mathcal{F}$ discussed in section 5, we use exponential-kernel linear regression models.

We compare against three baselines:

(i) **baseline-IS**: The standard non-robust method for maximizing the reward calculated using importance sampling policy evaluation on the training data and does not consider the shift,

(ii) **baseline-POEM** [Swaminathan and Joachims, 2015a] POEM improves the baseline-IS method by adding variance regularization to account for finite sample error, and

(iii) **baseline-DRO** [Si et al., 2020] as mentioned throughout the text, considers the shift over the joint (context, reward) distribution as opposed to separating them.

## 6.1 Evaluation Domains

We first provide a brief overview of our evaluation settings and provide full descriptions in the appendix. In one setting, the voting dataset setting, we consider the case when data from one city is used to train a policy deployed in some of the other cities. In the two other settings, to construct example shifts we cluster the datapoints by one of the context covariates into what we denote as *Coarse Groups* (cg). We then consider settings where the training distribution may consist of a different composition of cgs than the deployment/testing distribution. For each environment we give details in Table 1, such as the average reward on the training set of each (cg, action) pair.

**Simulated Setting:** Our simulated setting has 4 cgs. To mimic the non-linear relationships that may occur when using real data, unknown to the designer, cg 1 and 3 ($cg_{1,3}$) are identical, as are cgs 2 and 4 ($cg_{2,4}$). Each context is composed of the cg and 4 other values sampled from uniform distributions. For each of the 4 actions and cg, each (cg, action) pair has a set of true reward parameters $\phi^*$. The linear combination of $\phi^*$ and the contexts determine the Bernoulli parameter of the observed reward.

**Scene Setting:** We additionally test in a setting derived from the multiclass-supervised learning Scene classification dataset from the LibSVM [Chang and Lin, 2011] repository. We use a slightly modified version of the supervised to bandit conversion method from prior literature [Swaminathan and Joachims, 2015a, Agarwal et al., 2014]. In this method we assume we do not observe the correct label directly. Instead each label is an action and selecting the correct action/label gives positive reward, while the wrong action/label gives negative reward. We use the true label to create cgs and consider the case where we do not have access to the coarse group at test time so it cannot be used in learning the policy. This illustrates a more challenging and realistic case where there is shift in the unobserved context features. To create a context distribution shift between train and test, we sampled different proportions of datapoints from each group to put in the train distribution.

**Voting Setting:** The voting dataset by Gerber et al. [2008] contains data collected from a randomized controlled trial-style study testing various actions (1 control and 4 treatments) to increase voter turnout. We consider a potential real shift in the dataset by using data from one city to train a policy deployed in some of the other cities. To illustrate the worst case, we chose the training city and the deployment cities such that there is a large context difference between the cities (details in appendix). We consider the recorded binary voting outcome as our reward. Because the non-control actions require additional effort of printing and mailing a letter compared to the control (do nothing), we slightly lower the non-control action rewards by 0.09 to simulate this effort cost.

## 6.2 Results

We use the training dataset to optimize the policies and evaluate on the test dataset. For the voting dataset we use importance sampling to evaluate the policies. For Simulated and Scene, we have the true rewards for all action and use them for evaluation. All the robust algorithms have robustness parameters. FDRO has two, the reward distribution uncertainty set radius, $\delta_c$, the context distribution radius, $\delta_x$. We expect many of our settings predominantly have context shifts. We leverage this information to set our FDRO parameter for reward shift to $\delta_c = 0.001$. In Figure 1f and the appendix we give performance at different values of $\delta_c$ and show performance is similar for a range of values. We show our main results in the top row of Figure 1 where we consider the effect of $\delta_x$. Baseline-DRO has a single parameter for the KL-radius over the joint context-reward distribution ambiguity set, $\delta$. Baseline-POEM also has a single parameter, the variance regularization strength, $\lambda$. Faury et al. [2020] prove that POEM can be equivalently viewed from a DRO formulation. So for comparison when reporting the results, we first translate $\lambda$ to the corresponding DRO-radius $\delta = n\lambda^2$. We demonstrate in all three settings, FDRO is able to outperform all the baselines.

In Figure 1a, we consider the case of testing an algorithm fit with a fixed $\delta_x$ in different populations. We plot the performance gain of each algorithm over baseline-IS as the test population is increasingly different. We made this environment especially challenging for FDRO by fixing $\delta_x$ for FDRO while solving for the **best** $\delta$ for *each* different test population for baseline-DRO and baseline-POEM. We see that as expected when the test context distribution is nearly identical to the training context distribution, the robust policy learned by FDRO does not perform as well (negative reward difference). However as the test distribution differs, we see FDRO starts performing much better than the non-robust policy. Also as predicted, in setting with binary rewards, the joint formulation of baseline-DRO suffers from the same issues of degeneration in learning the optimal train-distribution policy as in

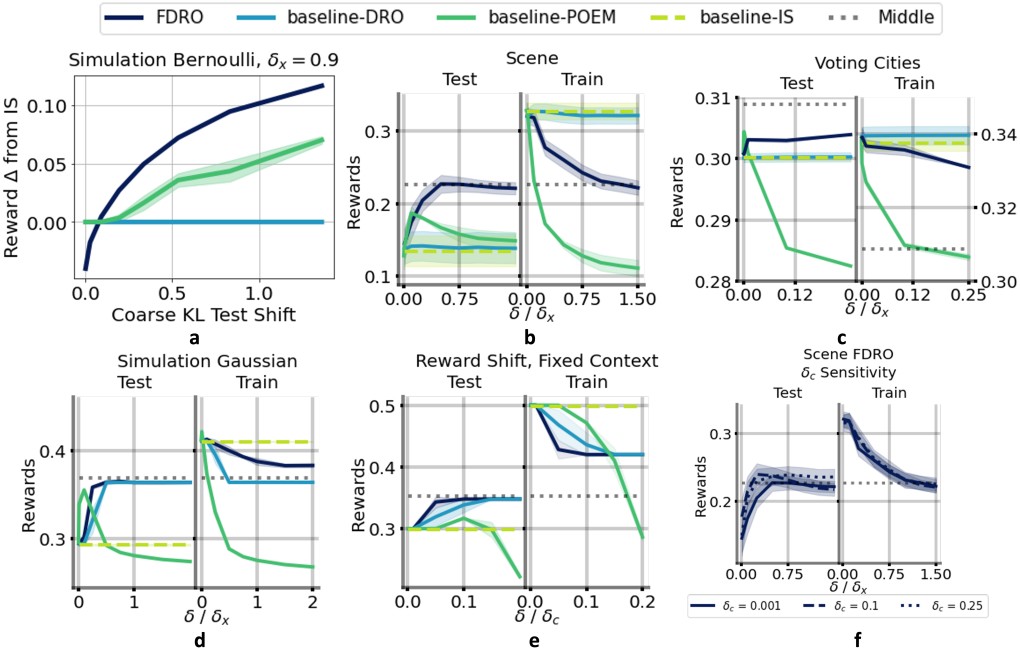

Figure 1: The dotted line 'Middle' serves as a constant horizontal reference across the two panels in plots (b-e). All plots are plotted with 95% confidence intervals shaded. Reported results and confidence intervals are averaged over 10 runs in all simulated domains, and 30 runs in Scene and Voting. **Top Row:** We plot our main results and demonstrate FDRO is able to learn more robust policies under context shift in all setting. (a) We vary the shift in the test distribution and plot results as the performance gain over baseline-IS. (b,c) We fix the train and test distributions and plot the performance at different $\delta/\delta_x$ hyperparameters in Scene and Voting. **Bottom Row:** We present discussion results: (d) Results from a continuous-reward domain: FDRO is able to outperform baselines (e) A domain where the test reward distribution shifts but context does not. (f) We show performance of FDRO at multiple values of $\delta_c$ for the Scene dataset.

the supervised DRO literature [Hu et al., 2018]. We additionally see that even with the advantage of using the best parameter per test population, baseline-POEM still does not perform as well.

In Figures 1b and c. We consider fixed training and test distributions and plot performance at different values of $\delta/\delta_x$ for the voting and scene settings. We see that with increasing $\delta_x$, FDRO can learn robust policies that perform well on the test set. Baseline-DRO suffers from the same issue previously discussed. The performance of baseline-POEM decreases on both distributions. As the variance regularization weight increases, POEM is incentivized to learn low variance policies, such as choosing low variance but low reward actions, or reducing the variance from importance sampling by choosing a policy close to the data-collection policy, which may be suboptimal. Figure 1c shows a real-world shift between cities, we can see if we would like to use data from one city to train a policy to deploy in other cities, it is important to consider distribution shift. Additionally for the case we consider with binary rewards it may be necessary to use FDRO.

### 6.3 Discussion

**Continuous Rewards and Reward Shift** We focus on the common bandit setting of binary rewards in the main result; however, we also show results for a continuous rewards setting in a simulations environment with Gaussian-distributioned rewards. In Figure 1d, we see that baseline-DRO is able to learn good polices, however FDRO with extra structure information was able to learn a better policy, with better train performance at the same test performance. Additionally, we explore reward shifts. We create a new simple environment (described in the appendix) where we can easily create a test environment with shifted rewards. We fix $\delta_x$ and sweep $\delta_c$ for FDRO and we see in Figure 1d, both FDRO and baseline-DRO are able to handle this this case well.

**When to use FDRO vs. baseline-DRO (equivalence of solutions)** One question is if the parameters can be set such that baseline-DRO and FDRO choose the same solution, and if not when one is beneficial over the other. To examine this question we can look at the KL-divergence chain rule decomposition:

$$D_{kl}(p_0(R, X)||p(R, X)) = D_{kl}(p_0(X)||p(X)) + E_{p_0(x)}[D_{kl}(p_0(R|X)||p(R|X))].$$

Recall $D_{kl}(p_0(R, X)||p(R, X)) = \delta$, $D_{kl}(p_0(X)||p(X)) = \delta_x$, $D_{kl}(p_0(R|X)||p(R|X)) = \delta_c$. We can see, when $\delta_c$ is the same for each $X$, we have $\delta = \delta_x + \delta_c$. When $\delta$ of baseline-DRO is set to $\delta_x + \delta_c$ baseline-DRO and FDRO consider distributions the same KL distance away, with FDRO placing additional constraints on the set of distributions searched. From the decomposition, we can see, with additional stipulations (details in appendix), we can set a family of $\delta_x$ and $\delta_c$ values for FDRO to achieve the same solution of baseline-DRO for each $\delta$ . However, for a given $\delta_x$ and $\delta_c$ it is not possible to find a $\delta$ for baseline-DRO that achieves the same solution. Baseline-DRO has less constraints and can, for example, place the shift "budget" ($\delta_x + \delta_c$) all in the rewards or all in the contexts if that is the worst case. This may cause the degenerate performance in the binary rewards case for baseline-DRO where reward shift dominates. FDRO can set limits on how much reward shift is considered and ensures context shift is considered, allows us to overcome this. This example informs us that generally, if the algorithm designer would like to ensure a certain amount of reward and context shift are considered, then it may be beneficial to use FDRO and invest the additional effort to decompose $\delta$ between $\delta_x$ and $\delta_c$.

**Setting the $\delta$ parameters and potential negative societal impacts** One limitation common to all DRO methods is that choosing the uncertainty set ($\delta$) may be difficult [Rahimian and Mehrotra, 2019]. Accounting for too large a shift may result in overly conservative policies. While DRO methods and FDRO are robust to shifts, if the shift is not as large as predicted, the policy learned using FDRO may be overly conservative which can potentially cause negative impact in terms of lower than necessary performance. To mitigate this risk, $\delta$ should be carefully set. Common approaches from prior work, which can apply to our method as well, is to use other similar datasets or sources to derive reasonable shifts. Additionally, empirically we see, our method often works for many values of $\delta$. For example in Figure 1a, we show while we may incur a small penalty by setting $\delta_x$ too large if the distribution does not shift, we can learn a policy that has much larger gains if the context distribution does shift. Similarly, in Figure 1f we show robustness to $\delta_c$. Another approach could be to utilize a human in the loop approach by examining the worst case distribution. The worst case distribution can be easily calculated for KL uncertainty sets and more details and demonstration are provided in the appendix.

## 7 Conclusion

In this work we consider the batch learning of contextual bandit policies when the deployment distribution may shift from that of data collection. We propose a distributionally robust optimization formulation that separates the reward and context shift. We demonstrated that our proposed estimators converge asymptotically and then developed a practical algorithm FDRO that is able to overcome some failure cases of prior methods and learn better performing robust policies.

## Acknowledgements

This work was supported in part by a Stanford Human-Centered AI Hoffman-Yee Grant.

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
