**Algorithm 1:** Double DRO policy optimization

**Data:** Dataset $\mathcal{D} = \{X_i, A_i, R_i\}$, $\pi_0$, $\delta_c$ (reward/cost KL radius), $\delta_x$ (context KL radius)

**Result:** $\hat{\pi}^*_{dro}, \hat{V}^*, \hat{\alpha}^*_x, \hat{\alpha}^*_c (n \times k$ values)

```
// First DRO (rewards) (this step only needs to be done once):
```
**1** **if** *Binning* **then**
**2**     $R_{X,A} \leftarrow$ Bin values of $R_i$ by unique $(X, A) \in \mathcal{X} \times \mathcal{A}$ pairs.;
**3**     Solve for $\hat{Q}_{X,A}$ for every unique $(X, A)$ using Eq. 5;
**4** **else**
```
     // Using function approximation
```
**5**     $\{\mathcal{F}\}_{i=1}^m \leftarrow$ Fit approximators to evaluate $\hat{\mathbb{E}}[\exp(-R/\alpha_c)]$ using the method in Sec. 5;
**6**     Solve for $\hat{Q}_{X_i,a}$ for every $X_i \in \mathcal{D}, a \in \mathcal{A}$ using Eq. 12 `// n×k total` $\hat{Q}_{X_i,a}$ `values`
```
// Policy Evaluation and Learning
```
**7** $\pi \leftarrow$ initial $\pi$;
**8** **while** $\pi$ *not converged* **do**
**9**     $J(\alpha_x) = \alpha_x \log \frac{1}{n} \sum_i \exp\left(\frac{-\sum_{a \in \mathcal{A}} \pi(a|X_i)\hat{Q}_{X_i,a}}{\alpha_x}\right) + \alpha_x \delta_x$;
**10**    $\hat{V}_\pi = -\min_{\alpha_x \geq 0} J(\alpha_x)$, $\hat{\alpha}^*_x = \operatorname{argmin}_{\alpha_x \geq 0} J(\alpha_x)$ `// Policy Evaluation`
**11**    Update($\pi$) (ex. for differentiable $\pi$, fix $\hat{\alpha}^*_x$ and perform gradient ascent.)
```
     // Note the optimization problem on lines 3, 6, and 10 are all convex
        and can be solved using gradient descent, bisection search, etc.
```

# A   Algorithm and Additional Experimental Details

## A.1   Complete Algorithm

We give the pseudocode of our algorithm in Algorithm 1

## A.2   Experiment Details

In this section we give additional details for our experiments not included in the main text. Each experiment is run on an internal cluster machine with 4CPU and 16GB of memory. A single run of each experiment runs between 10 minutes and 2 hours depending on dataset size.

For all experiments, our $\phi$ vector is defined as:

$$\phi(X, A) = [X, OneHot(A)] \tag{14}$$

### A.2.1   Simulated Experiments

As mentioned in the main text, our simulated setting has 4 cgs, where cgs 1 and 3 ($cg_{1,3}$), and cgs 2 and 4, ($cg_{2,4}$) are the same. In addition to the cgs, there is also 4 other context features, $X_i^f$ randomly sampled from a uniform distribution. The first 2 features are randomly sampled from Uniform($[0, 0.4]$) while the second 2 features are randomly sampled from Uniform($[0.3, 0.7]$). There are 4 actions and each $(cg, a)$ pair has a set of true reward generating parameters $\phi^*_{(1,3),a}$ and $\phi^*_{(2,4),a}$ of length 4, and the observed reward is $R_i = $ Bernoulli($\phi^{*T}_{X_i,a} X_i^f$). The $\phi^*$'s are:

$$\phi^*_{(1,3),a} = \begin{cases} a_0: & [0.5, 0.0, 0.5, 0.0] \\ a_1: & [1.0, 0.0, 0.0, 0.0] \\ a_2: & [0.0, 0.0, 1.0, 0.0] \\ a_3: & [0.1, 0.1, 0.1, 0.1] \end{cases}, \quad \phi^*_{(2,4),a} = \begin{cases} a_0: & [0.5, 0.0, 0.5, 0.0] \\ a_1: & [0.0, 0.0, 0.0, 1.0] \\ a_2: & [0.0, 1.0, 0.0, 0.0] \\ a_3: & [0.1, 0.1, 0.1, 0.1] \end{cases}$$

We use a misspecified context for the experiments and the full context used is the $cg$ concatenated with the other features $X_i = [cg, X_i^f]$.

For the Gaussian experiments, we use the same setup, but instead we first add gaussian noise to the rewards and then clip them to $[0,1]$: $R_i = \text{clip}(\phi_{X_i,a}^{*T} X_i^f + \mathcal{N}(0,0.1), \min = 0, \max = 1)$. For the plot in the main text, the training dataset population split is as listed in Table 1 while the test dataset population split between the coarse groups was shifted to $(0.4, 0.1, 0.4, 01)$.

Reported results are averaged over 10 runs where we average over the randomization of the dataset split, the initial policy parameters, and the gradient descent procedure.

### A.2.2 Voting Experiments

The voting dataset of Gerber et al. [2008] is licensed under the Creative Commons Attribution-Non Commercial-No Derivatives 3.0 license. The dataset contains 180002 datapoints, each a voter in a different household across the state of Michigan. The researchers designed one control, and 4 treatment actions that involved mailing the selected individuals a letter ahead of the 2006 Michigan primary election. The actions are:

- Nothing (control): No letter is sent
- Civic: A letter with "Do your civic duty"
- Hawthorne: A letter with "You are being studied"
- Self-History: A letter with the voter's past voting participation record as well as that of the other members of the household. The letter also mentioned a follow-up letter will be sent with the household's updated voting participation after the election
- Neighbors-History: A letter with the voting participation records of the individual, the other members of the individual's household, as well as the neighbors. The letter also mentioned a follow-up letter will be sent after the election with everyone's updated participation, so the individual's participation will be made known among the neighbors.

The data collection policy randomly sampled actions with probability $\frac{5}{9}$ for the control action and $\frac{1}{9}$ for the other actions. We use the binary indicator of voting outcomes as the reward. The neighbors action was the best action for the population, with participants under that action voting at a rate of around $10\%$ higher than those under the control action. However all actions except the control require effort and cost in the form of printing and mailing a letter. Therefore, for each non-control action, we induce an "effort" cost by randomly flipping $0.09n_a$ of the positive outcomes to 0, where $n_a$ for each action is the number of participants that experienced that action. This lowers the average reward for all non-control action by 0.09. We choose this cost to induce heterogeneous effects within the dataset.

For the experiment in the main text we take data from one city to use to train a policy deployed on other cities. We choose the train and deployment cities to have a large context shift. For simplicity we approximate a large context shift by choosing one context feature to examine, the 'p2004' feature which is an indicator if the individual voted in the 2004 primary election. We chose the one city with the largest average 'p2004' to be the training city and the 35 cities with the smallest 'p2004' to be the test cities. For computation efficiency and speed, we use a random sample of $25\%$ of the training data.

We additionally provide a experiment similar to that of the artificial shift in the Scene and Simulated datasets that uses all the data. We set the 'coarse group' using this p2004 features. From the subgroup that did vote in the 2004 primary election, we partition 0.8 of the dataset into the training set and the remaining 0.2 into the test set. Of the subgroup that did not vote, we partition 0.2 of the dataset into the training set and the remaining 0.8 into the test set. This gives the coarse group composition of $(0.75, 0.25)$ in the training set and $(0.15, 0.85)$ in the test set. While we choose this split, different splits where there is a shift between the training distribution and test distribution also work. We use the features of household size and gender as contexts. We illustrate the more difficult case where some variables, such as the coarse group variable (whether the participants voted in the 2004 primaries), are not available at test time.

We report results averaged across 30 runs.

### A.2.3 Scene Experiments

The Scene classification dataset from the LibSVM library Chang and Lin [2011] is a multiclass-supervised learning dataset. Datsets in the LibSVM library are licensed under the BSD 3-Clause. The

features are processed features of the image and the labels correspond to various landscapes (such as 'mountains', etc). As mentioned in the main text, we use a supervised-to-bandit conversion method where we assume we do not observe the correct labels directly and instead each label is an action and selecting a correct action/label gives positive reward (+1) while a wrong action/label gives no reward (0). We assume we only observe 1 action per datapoint and we randomly sample the actions we observe. We use the true labels to partition the dataset into coarse groups, we place datapoints that contain labels 1, 2, and 3 into one group ($cg_0$) and the others into another group ($cg_1$). As in the voting dataset we induce a train test split such that 0.1 of the datapoints in $cg_0$ and 0.6 of the datapoints in $cg_1$ are in the train dataset with the rest in the test dataset. As in the voting dataset, we assume this feature will not be available at test time.

Due the small dataset size, we use orthonormal random projection, $P \in \mathbb{R}^{d \times d_p}$ to project the dataset onto 3 dimensions. Let $d = 294$ be the original context feature length and $d_p = 3$ be the desired projected length. To create our random orthonormal projection matrix of size $\mathbb{R}^{d \times d_p}$ we first sample a matrix of size $\mathbb{R}^{d \times d_p}$ where each item is sampled from Uniform($[0, 1]$). We then take the singular value decomposition and calculate the left and right singular values $U \in \mathbb{R}^{d \times d_p}$ and $V \in \mathbb{R}^{d_p \times d_p}$. We then calculate: $P = UV$.

Our reported results are averaged over 30 runs where each run randomizes across the datapoints in the train-test splits, the initial policy parameters, the gradient descent steps during policy optimization, and the random generated random projection matrix.

### A.2.4 Reward Shift Experiment

Our simple environment for reward shift has two unique contexts ($X_0$, $X_1$), with $X_0 = [0, 1]$ and $X_1 = [1, 0]$, and 3 actions. We assume the rewards are binary, but with different support for each ($X$, $a$) pair. The reward generation process is defined by a tuple of 3 values ($r_0, r_1, p$) with $r_1 \geq r_0$ and $R_i = (r_1 - r_0)$Bernoulli(p) $+ r_0$.

$$X_0 = \begin{cases} a_0 : & (0.0, 1.0, p) \\ a_1 : & (0.3, 0.5, p) \\ a_2 : & (0.1, 0.15, p) \end{cases} , \quad X_1 = \begin{cases} a_0 : & (0.0, 1.0, p) \\ a_1 : & (0.0, 1.0, p) \\ a_2 : & (0.1, 0.15, p) \end{cases}$$

We induce a reward shift by setting $p = 0.7$ when generating the training data and $p = 0.5$ for the test data.

### A.3 Additional Results: Different dataset splits and hyperparameters

In Figure 2 we give some results from different dataset splits and hyperparameters. In Figure 2 (a) we give the plot similar to plots (b-e) in Figure 1 of the main text for the Simulated Bernoulli experiment. In these plots, we fix the training and test distribution and examine different values of the $\delta/\delta_x$ hyperparameter. We see FDRO can learn more robust policies that work well in the test set under context shift. In Figure 2(b) and (c), we show the result of running different dataset splits. Figure 2(b) considers a different training and test context distributions. It considers a shift of group distribution from $(0.63, 0.17)$ in the training set to $(0.3, 0.7)$ in the test set. Figure 2(b) considers a different number of groups. As opposed to 2 coarse groups, we now partition the dataset into 3 groups (with items with labels 0 and 1 in $cg_0$, the remaining items with labels 2, 3 in $cg_1$, and all the remaining items in $cg_2$). We see in both cases FDRO is able to learn more robust policies that work well on the test context distribution compared to baselines. In Figure 2(d) and (e) we test out different hyperparameters to generate plots similar to that of Figure 1(a) of the main text. Like in the main text plot, we fix the $\delta_x$ and training distribution and generate test datasets with different shifted group distributions. Also, as in the main text, while we fix $\delta_x$ for FDRO, we optimize the uncertainty parameter at each test distribution for the baselines. We plot the performance of each algorithm as the performance gain in the test dataset over baseline-IS. We see that for different $\delta_x$ hyperparameters, FDRO can still perform better than baselines when there is shift. Setting a lower value of $\delta_x$ leads to a smaller decrease in performance when there is very little shift; however, the benefits at large shifts is also smaller.

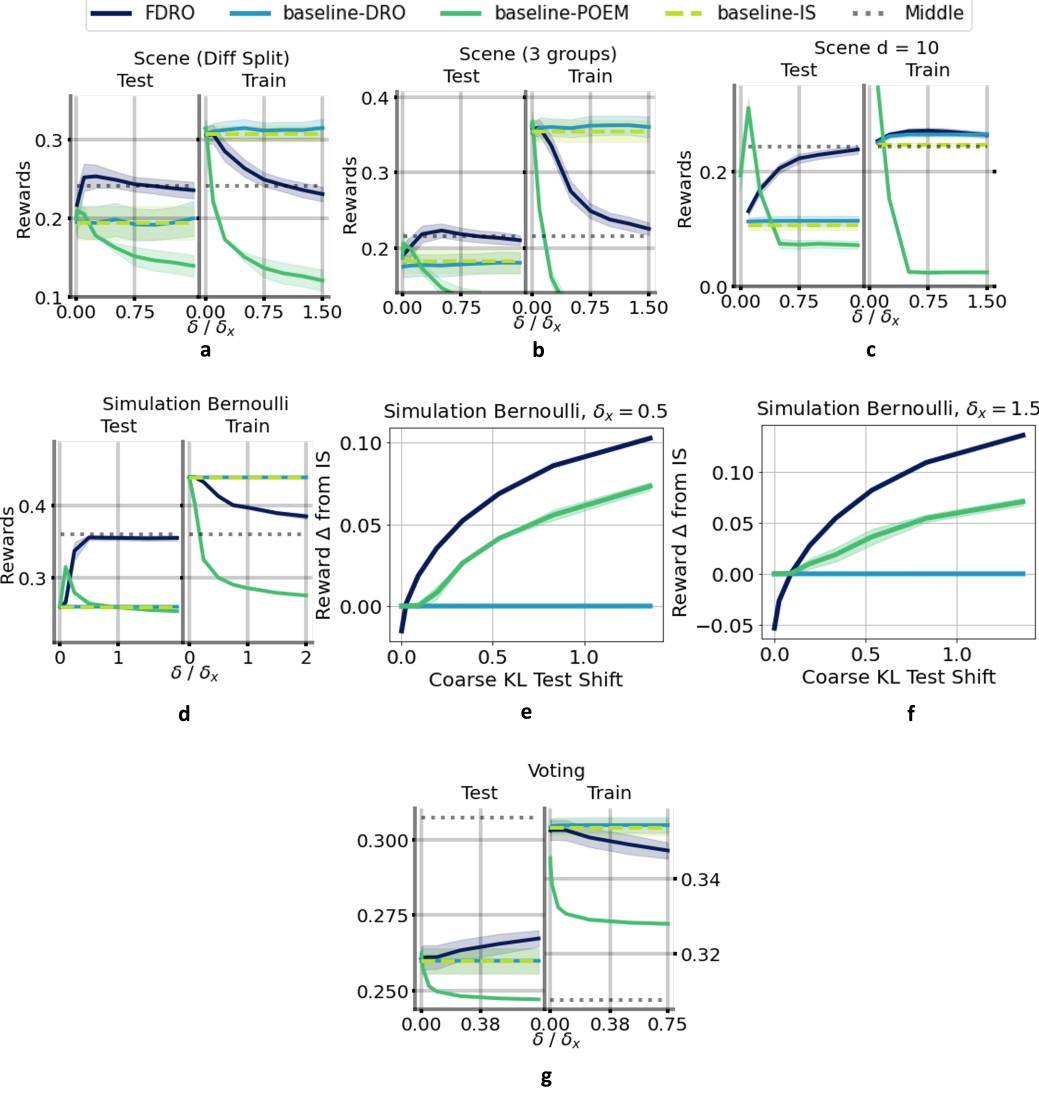

Figure 2: In these plots, we demonstrate FDRO can perform well even under different hyperparameter and dataset configurations. The dotted line 'Middle' serves as a constant horizontal reference across the two panels in plots (a-d, g). All plots are plotted with 95% confidence intervals shaded. Reported results and confidence intervals are averaged over 10 runs in all simulated domains and 30 runs in Scene and Voting. **Top Row:** (a-c) We fix the train and test distributions and plot the performance at different $\delta/\delta_x$ hyperparameters in the Scene domain. The plots show results from different configurations than that in the main text, including different dataset splits (a), different number of groups (b), and different number of features for random projection (c). We see FDRO is able to learn more robust policies that does well in the test distribution in all cases. **Middle Row:** (d-f) We give results for different configurations in the Simulated Bernoulli dataset. In (d) we give the two panel plot with the parameters described in the main text. In (e,f) We vary the shift in the test distribution and plot results as the performance gain over baseline-IS. **Bottom Row:** We give a plot for the voting dataset with an artifically induced shift.

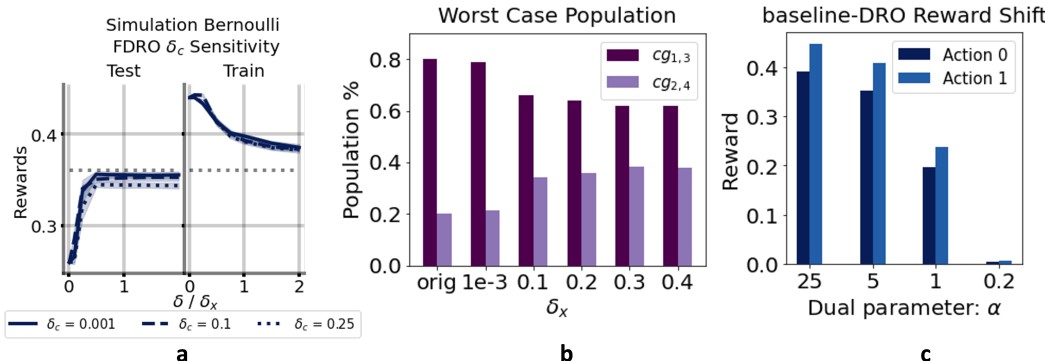

Figure 3: We give additional results on hyperparameter sensitivity. The dotted line 'Middle' serves as a constant horizontal reference across the two panels in plots (a) we show sensitivity to $\delta_c$ by showing performance Simulation Bernoulli at multiple values of $\delta_c$ (b) The extracted worst case population of each $cg$ of our FDRO for different $\delta_x$. (c) We plot the reward for each action for the joint formulation (baseline-DRO) at different values of the dual parameter $\alpha$. As described in the text, the order between the actions never changes.

## A.4   Additional Discussion

We provide some additional points of discussion. In Figure 3a we plot performance at different values of $\delta_c$ for the Simulated Bernoulli case and show performance to be similar across parameters.

### A.4.1   Examining the Worst Case Context

The worst case distribution for DRO formulation with KL distance bounds is an softmax scaled weighting of the datapoints [Hu and Hong, 2013]. with a temperature determined by optimal dual variable $\alpha^*$. If we have discrete or easily discretized context features, we can examine the worst case distribution of that feature. We plot the worst case distributions of the $cg$ context feature of FDRO in Figure 3b in the Bernoulli simulated environment. For each $\delta$ value, the end user can examine the estimated performance on the training distribution as well as the worst-case distribution of some of the features to decide if the policy is reasonable.

Formally, from [Hu et al. [2018]], the worst case distribution is an exponentially/softmax reweighing of the points:

$$p_i = \frac{\exp((-\sum_{a \in \mathcal{A}} \hat{\pi}^*(a|X_i)\hat{Q}_{X,A})/\alpha_x^*)}{\sum_i \exp((-\sum_{a \in \mathcal{A}} \hat{\pi}^*(a|X_i)\hat{Q}_{X,A})/\alpha_x^*)} \tag{15}$$

where $\alpha_x^*$ is the optimal dual variable of $\hat{\pi}^*$ in Equation 7. Consider the $j^{th}$ feature in the context and assume this feature is either discrete, or easily discretized. We can calculate the worst case distribution over values the feature takes on as:

$$P(X[j] = x) = \sum p_i \mathbb{1}[X_i[j] = x] \tag{16}$$

### A.4.2   Utilizing unavailable features:

In two of the settings - voting and scene, we assumed there were context features in the train set that are unavailable at deployment. This situation occurs when those features are unavailable at deployment, or when they are sensitive features that should not be used for fairness considerations. Even without using it for the policy, FDRO is still able to better utilize this information in the first DRO step when calculating the worst case rewards $\hat{Q}_{X,A}$.

### A.4.3   Additional Discussion: Equivalence of Solutions

As mentioned in the main text, we can compare baseline-DRO and FDRO by examining the KL-divergence chain rule decomposition:

$$D_{kl}(p_0(R,X)||p(R,X)) = D_{kl}(p_0(X)||p(X)) + E_{P_{x_0}}[D_{kl}(p_0(R|X)||p(R|X))]$$

Recall $D_{kl}(p_0(R,X)||p(R,X)) = \delta$, $D_{kl}(p_0(X)||p(X)) = \delta_x$, $D_{kl}(p_0(R|X)||p(R|X)) = \delta_c$. This informs us that we have the following relation between the baseline-DRO parameter $\delta$ and the FDRO ones $\delta_x$ and $\delta_c$:

$$\delta = \delta_x + \sum_x P_{x_0} \delta_c$$

.

From this we can see that if we allow $\delta_c$ to vary with context $X$ and action $A$, we can set parameters of $\delta_x$ and $\delta_c(X, A)$ such that the FDRO will find the same solution as baseline-DRO. However this increase in parameter may be difficult for the algorithm designer. Therefore we expect FRDO to be particularly useful when there are independent shifts in the reward and context distributions, which is a common setting– such as shifting locations (for hospitals or social welfare solutions).

We give a detailed example of what the worst case shift in rewards look like for a single action, and how differs between the factored formulation (FDRO) and the joint formulation (baseline-DRO) Figure 4. In this example, we consider a simple two context, discrete reward setting and examine the worst case distributions for a single action. We compare with setting the single joint parameter to $\delta = 0.1$ and setting the context shift parameter of the factored formulation to $\delta_x = 0.1$, while allowing for very little reward shift $\delta_c = 0$. We can see the resulting worst case joint distribution is different under the two formulations. We can additionally see for the factored formulation to achieve the equivalent worst case distribution as the joint formulation, we need to set different reward shift radius parameters ($\delta_c$) for context $X_0$ and context $X_1$.

### A.4.4 Additional Details: Why Joint DRO May Not Work for Binary Rewards

In the main text we mentioned the formulation from prior work [Si et al., 2020] that jointly considers the context and reward distribution shift may lead to a degenerate result of finding the best policy on the training context distribution under binary reward feedback. This empirical observation is similar to a proven phenomena from the supervised learning literature in distributionally robust optimization [Hu et al., 2018] under continuous inputs. The intuition behind this phenomena is that DRO can be seen as weighing the datapoints based on the incurred loss/or incurred cost, where cost are negative rewards. Under binary reward feedback, the lower reward points for each context action pair are all upweighted equally and we empirically observe this leads to a learned policy equivalent to that optimal for the training context distribution. In this section we provide a simple example and additional details around this observation.

Consider a simplified version of the Simulated Bernoulli setting described above, however in this setting, let $cg$ be the only context feature. Additionally in this case there are only 2 actions $\{a_0, a_1\}$. $a_0$ achieves medium reward for all contexts (Bernoulli(0.4)), while $a_1$ achieves high reward for $cg_{1,3}$ (Bernoulli(0.5)) and low reward for $cg_{2,4}$ (Bernoulli(0.3)). Consider optimizing for the parameters, $\theta$, of a linear contextual bandit policy of the form $\operatorname{argmax}_a \phi(x,a)^T \theta$ where $\phi(x,a)$ concatenates the context with a one hot encoding of the action. This representation attempts to fit a linear dependence on context when the relationship is very nonlinear. Due to this misspecification, this ends up being equivalent to optimizing a non-contextual bandit which chooses the single best action. Consider the case where $cg_{1,3}$ comprises a large majority (0.8) of the training context distribution, $P_{x,train}$. In this case, action $a_0$ is optimal for $P_{x,train}$. If we are certain only the context distribution will shift, as we expand the uncertainty set around $P_{x,train}$, we expect best worst-case action to eventually shift to the action that is good for both groups, $a_1$. However due to binary rewards, we empirically see the joint formulation upweights the lower rewards for each action equally. We demonstrate this in Figure 3(c), where we examine the re-weighted rewards for each action in the joint formulation, which the joint method uses with weighted importance sampling to optimize the policy. We look at the reward values under different values of the dual variable $\alpha$. Notice that for decreasing $\alpha$, while the re-weighted rewards are lower, the order of the benefit of the rewards does not change. Therefore in this example, there is no value of $\delta$ that will result in a policy different from that optimal for the training context distribution. Our method which decomposes context and reward generation shift mitigates this issue.

## B   Proofs

Throughout the proofs, we will define a new variable $C$, or costs, for convenience, with $C = -R$ and $C_i = -R_i$ for every $i$.

**Simple 2 Context, Discrete Reward Case**

**Data Collection Distribution**

Two Contexts

$X_0$
$X_1$

Context Distribution

$P_0(X_0) = 0.8$
$P_0(X_1) = 0.2$

Action 0 ($A_0$), Conditional Distribution
$P_0(\text{reward} | \text{context}, A_0)$

| reward | 1.0 | 0.8 | 0.4 | 0.0 |
|--------|-----|-----|-----|-----|
| $X_0$ | 0.8 | - | 0.2 | - |
| $X_1$ | - | 0.5 | - | 0.5 |

Expected Reward

$E[r | X_0, A_0] = 0.88$
$E[r | X_1, A_0] = 0.40$

Action 0 ($A_0$), Joint Distribution
$P_0(\text{Reward, Context} | A_0)$

| reward | 1.0 | 0.8 | 0.4 | 0.0 |
|--------|-----|-----|-----|-----|
| $X_0$ | 0.64 | - | 0.16 | - |
| $X_1$ | - | 0.1 | - | 0.1 |

$E[r | A_0] = 0.784$

Worst Case
Factored Shift:
$\delta_x = 0.1$
$\delta_c = $ Almost 0

**Factored-Shifted Distribution**

Context Distribution

$P_{shift}(X_0) = 0.6$
$P_{shift}(X_1) = 0.4$

Action 0 ($A_0$), Conditional Distribution
$P(\text{reward} | \text{context}, A_0)$ (no shift due to $\delta_c$)

| reward | 1.0 | 0.8 | 0.4 | 0.0 |
|--------|-----|-----|-----|-----|
| $X_0$ | 0.8 | - | 0.2 | - |
| $X_1$ | - | 0.5 | - | 0.5 |

Expected Reward

$E[r | X_0, A_0] = 0.88$
$E[r | X_1, A_0] = 0.40$

Action 0 ($A_0$), Joint Distribution
$P_0(\text{Reward, Context} | A_0)$

| reward | 1.0 | 0.8 | 0.4 | 0.0 |
|--------|-----|-----|-----|-----|
| $X_0$ | 0.48 | - | 0.18 | - |
| $X_1$ | - | 0.2 | - | 0.2 |

$E[r | A_0] = 0.688$

Worst Case
Joint Shift:
$\delta = 0.1$

**Joint Shifted Distribution (Baseline-DRO)**

Action 0 ($A_0$), Joint Distribution
$P_{shift}(\text{Reward, Context} | A_0)$

| reward | 1.0 | 0.8 | 0.4 | 0.0 |
|--------|-----|-----|-----|-----|
| $X_0$ | 0.46 | - | 0.22 | - |
| $X_1$ | - | 0.1 | - | 0.22 |

Context Distribution

$P_{shift}(X_0) = 0.68$
$P_{shift}(X_1) = 0.32$

Action 0 ($A_0$), Conditional Distribution
$P(\text{Reward} | \text{Context}, A_0)$

| reward | 1.0 | 0.8 | 0.4 | 0.0 |
|--------|-----|-----|-----|-----|
| $X_0$ | 0.68 | - | 0.32 | - |
| $X_1$ | - | 0.31 | - | 0.69 |

Equivalent
Context KL Shift:
$\delta_x = 0.04$

Equivalent Reward KL Shifts:
$\delta_{c,0} = 0.04$
$\delta_{c,1} = 0.07$

$\delta \sim = \delta_x + 0.68\, \delta_{c,0} + 0.32\, \delta_{c,1}$

Expected Reward (shifted)

$E[r | X_0, A_0] = 0.81$
$E[r | X_1, A_0] = 0.24$

$E[r | A_0] = 0.628$

Figure 4: A detailed example of what the worst case shift in rewards look like for a single action. We choose a simple two context, discrete reward setting. We see that even under the same amount of KL shift ($\delta = 0.1$), the difference in results under the factored approach and the joint formulations are different. We also give the equivalent parameters for FDRO to be equivalent to the solutions of the joint formulation.

## B.1 The Convexity and Derivatives of the Policy Evaluation steps

We first prove the convexity of the objectives inside the minimization of equations 6 and 7 and give their derivatives as mentioned in the main text. We first give this intermediate result:

**Lemma 1** (Convexity of DRO formulation). *Consider a set of $n$ values, $\{B_i\}$ for $i = 1, ..., n$ and function*

$$f(\alpha) = \alpha \log \left( \sum_i \frac{1}{n} \exp \left( \frac{B_i}{\alpha} \right) \right) + \alpha \delta$$

*Then $f(\alpha)$ is convex in $\alpha$ for $\alpha > 0$*

*Proof.*

$$f(\alpha) = \alpha \log \left( \sum_i \frac{1}{n} \exp \left( \frac{B_i}{\alpha} \right) \right) + \alpha \delta$$

$$= \alpha \log \left( \sum_i \exp \left( \frac{B_i}{\alpha} \right) \right) + \alpha \delta + \alpha \log \left( \frac{1}{n} \right)$$

First recall a classic result from convex optimization that states a function $f(x, y) = y g(\frac{x}{y})$ is convex for $y > 0$ if an only if $g$ is convex. [Boyd et al., 2004]

Note the expression $g(B) = \log \left( \sum_i \exp (B_i) \right)$ takes on the form of a log-sum-exp which is convex. Therefore the function $f(B, \alpha) = \alpha g \left( \frac{B}{\alpha} \right) = \alpha \log \sum_i \left( \exp \left( \frac{B_i}{\alpha} \right) \right)$ is convex, and consequently convex in $\alpha$ when $\alpha > 0$.

The linear combination of convex function is still convex, so we have the desired result. $\square$

Now we return to proving the convexity of equations 6 and 7. We first reproduce them here with new intermediate variables $\hat{Z}_{X,A}$ and $\hat{J}(\pi, \alpha_x)$ that represent the objective inside the optimization:

$$\hat{Z}_{X,A}(\alpha_c) = \alpha_c \log \left( \frac{1}{m_{xa}} \sum_{R_j \in \mathcal{R}_{X,A}} \left[ \exp \left( \frac{-R_j}{\alpha_c} \right) \right] \right) + \alpha_c \delta_c \tag{17}$$

$$\hat{J}(\pi, \alpha_x) = \alpha_x \log \sum_i \frac{1}{n} \exp \left( \frac{-\sum_{a \in \mathcal{A}} \pi(a|X_i) \hat{Q}_{X_i, a}}{\alpha_x} \right) + \alpha_x \delta_x \tag{18}$$

With these variables, we have:

$$\hat{Q}_{X,A} = -\min_{\alpha_c \geq 0} \left\{ \hat{Z}_{X,A}(\alpha_c) \right\} \quad \text{and} \quad \hat{V}(\pi) = -\min_{\alpha_x \geq 0} \left\{ \hat{J}(\pi, \alpha_x) \right\} \tag{19}$$

We can directly apply Lemma 1 to $\hat{Z}_{X,A}(\alpha_c)$ and $\hat{J}(\pi, \alpha_x)$ as defined above with $B_j = R_j$ and $B_j = \sum_{a \in \mathcal{A}} \pi(a|X_j) \hat{Q}_{X_j, a}$ respectively to obtain both $\hat{Z}_{X,A}(\alpha_c)$ and $\hat{J}(\pi, \alpha_x)$ are convex.

We also give the gradient of $\hat{Z}_{X,A}(\alpha_c)$ and $\hat{J}(\pi, \alpha_x)$:

**Remark 1** (Gradient of DRO formulation). *The gradient $\frac{\partial f(\alpha)}{\partial \alpha}$ of $f(\alpha)$ given in Lemma 1 is:*

$$\frac{\partial f(\alpha)}{\partial \alpha} = -\frac{\sum_i B_i \exp \left( \frac{B_i}{\alpha} \right)}{\alpha \sum_i \exp \left( \frac{B_i}{\alpha} \right)} + \log \left( \sum_i \exp \left( \frac{B_i}{\alpha} \right) \right) + \delta$$

We now prove the main lemmas and theorems from the main text.

## B.2 Collection of Useful Results From Prior Work

We first collect results from prior work we will use for proofs of Theorems 1, 2 and 3.

We will utilize Hoeffding's inequality:

**Lemma 2** (Hoeffding's Inequality). *Let $Z_1, ... Z_n$ be independent bounded random variables with $Z_i \in [a, b]$ for all i, where $-\infty < a \leq b < \infty$. Then:*

$$\mathbb{P}\left(\frac{1}{n}\sum_{i=1}^{n}(Z_i - \mathbb{E}[Z_i]) \geq t\right) \leq \exp\left(-\frac{2nt^2}{(b-a)^2}\right) \tag{20}$$

*and*

$$\mathbb{P}\left(\frac{1}{n}\sum_{i=1}^{n}(Z_i - \mathbb{E}[Z_i]) \leq t\right) \leq \exp\left(-\frac{2nt^2}{(b-a)^2}\right) \tag{21}$$

*for all $t \geq 0$*

### Distributionally Robust Optimization with KL divergence [Hu and Hong, 2013]

We also use a result from the DRO literature in stochastic optimization from Hu and Hong [2013]. The distributionally robust optimization formulation from stochastic optimization literature considers optimizing the model variables $\theta \in \Theta$ (for example, parameters of a logisitic regression model) to achieve the best worst-case future performance (ex. performance on the test set). The goal is to use the observed random values $\xi$ (ex. (input, label) pairs of the training data) to optimize a cost function $\mathcal{L}(\theta, \xi)$ (ex.classification loss) when evaluated in the unknown test distribution. The test distribution is assumed to be within an uncertainty set, $\mathbb{P}_{\xi_0}$. In this work, we focus on ambiguity sets defined as a bound on the Kullback-Leibler (KL) divergence from the data generation distribution $P_{\xi_0}$. The DRO optimization problem is:

$$\theta^* = \arg\min_{\theta \in \Theta} \max_{P_\xi \in \mathbb{P}_\xi} \mathbb{E}_{P_\xi}[\mathcal{L}(\theta, \xi)], \text{ where } \mathbb{P}_\xi = \{P_\xi : D_{kl}(P_\xi || P_0) \leq \delta\}$$

The inner maximization over $P_\xi$ is difficult to solve directly as it is a constrained optimization problem in probability space. However Hu and Hong [2013] show through strong duality that this can be transformed into a much easier unconstrained convex minimization over a scalar:

**Lemma 3** (Theorem 1 from Hu and Hong [2013]:). *Assume $\mathcal{L}(\theta, \xi)$ takes on a finite set of values or is bounded almost surely. Then through strong duality, the inner maximization is equivalent to solving the following optimization problem:*

$$\max_{P_\xi \in \mathbb{P}_\xi} \mathbb{E}_{P_\xi}[\mathcal{L}(\theta, \xi)] = \min_{\alpha \geq 0} \alpha \log \mathbb{E}_{P_0}[\exp(\mathcal{L}(\theta, \xi)/\alpha)] + \alpha\delta$$

### Uniform Convergence of Stochastic Optimization [Shalev-Shwartz et al., 2009]

We will also make use of the following uniform convergence result from Shalev-Shwartz et al. [2009] of the optimization of convex objectives.

Consider the following stochastic minimization problem:

$$w^* = \underset{w \in \mathcal{W}}{\operatorname{argmin}} F(w) \tag{22}$$

Where $f(w; Z)$ denote the loss objective and $F(w) = \mathbb{E}_Z[f(w, Z)]$. Define the empirical objective and minimizer as $\hat{F}(w) = \frac{1}{n}\sum_{i=1}^{n} f(w; z_i)$ and $\hat{w} = \operatorname{argmin}_w \hat{F}(w)$.

Then the following result guarantees uniform convergence of $F(w)$ over $w$:

**Lemma 4** (Uniform Convergence of Stochastic Convex Optimization (Theorem 5 of Shalev-Shwartz et al. [2009])). *Let the following conditions hold:*

1. *$f(w, z)$ is convex in $w$*

2. *$\mathcal{W} \subset \mathbb{R}^d$ is bounded by $B$*

3. $f(w, z)$ *is L-Lipschitz w.r.t $w$*

*Then with probability of at least $1 - \delta$, for all $w \in \mathcal{W}$:*

$$\left| F(w) - \hat{F}(w) \right| \leq \mathcal{O}\left( LB\sqrt{\frac{d \log(n) \log(d/\delta)}{n}} \right) \tag{23}$$

We also use Si et al. [2020]'s result on the upper bound on $\alpha$:

**Lemma 5** (Lemma 11 of Si et al. [2020]). *Let variable B be bounded between 0 and M, then the optimal $\alpha$ in the objective:*

$$\inf_{\alpha \geq 0} \alpha \log \mathbb{E}_{P_B}[\exp(B/\alpha)] + \alpha\delta$$

*is upper bounded: $\alpha^* \leq M/\delta$.*

## B.3 Proof of Theorem 1

We first restate Theorem 1:

**Theorem 1** (Strong Duality). *The optimization problem in equation 2 is equivalent to solving:*

$$V(\pi) = -\min_{\alpha_x > 0} \left\{ \alpha_x \log \mathbb{E}_{X \sim P_{x_0}} \left[ \exp\left( \frac{\mathbb{E}_\pi[-Q_{X,A}]}{\alpha_x} \right) + \alpha_x \delta_x \right] \right\}, \tag{4}$$

*where* $Q_{X,A} = -\min_{\alpha_c > 0} \left\{ \alpha_c \log \mathbb{E}_{R \sim P_{r_0|X,A}} \left[ \exp\left( \frac{-R}{\alpha_c} \right) \right] + \alpha_c \delta_c \right\}.$ \tag{5}

The proof of Theorem 1 directly follows from applying Lemma 3 twice. Notice that because rewards, R, are bounded $[0, R_M]$, the assumption of Lemma 3 is satisfied for $Q_{X,A}$. Similarly, $Q_{X,A}$ are also bounded $[0, R_M]$ so the assumption is also satisfied for $V(\pi)$.

## B.4 Policy Evaluation and Learning convergence

We first prove the probability of convergence for $\hat{Q}_{X,A}$ for each $(X, A)$ pair separately. We then apply the union bound to bound the probability they hold jointly for all $(X, A)$ pairs. We then use this result along with local Lipschitz continuity to prove the rest of the results.

We start with proving PAC uniform convergence for $\hat{Q}_{X,A}$:

**Lemma 6** (PAC convergence of $Q_{X,A}$ (first DRO layer)). *Given a $(X, A)$ pair, when $n \geq \frac{(p_0(x,a))^2}{2\log(2/\delta)}$, the following result holds with probability at least $1 - \delta$ and all $\alpha_c \in [\underline{\alpha}_c/2, 2\bar{\alpha}_c]$:*

$$\left| Q_{X,A} - \hat{Q}_{X,A} \right| \leq \mathcal{O}\left( c\sqrt{\frac{\log(n) \log(2/\delta)}{n}} \right) \tag{24}$$

*where*

$$c = \frac{32(R_{max})^3}{\delta_c^2 \underline{\alpha}_c^2 \exp(-R_{max}/\underline{\alpha}_c) \underline{p}_0(x,a)} \tag{25}$$

*and $\underline{p}_0(x, a)$ is the lowest probability $(X, A)$ pair: $\underline{p}_0(x, a) = \min_{(x,a)} p_0(x, a)$, and $\underline{\alpha}_c$ is the minimum value for $\alpha_c$ and satisfies $\underline{\alpha}_c > 0$.*

*Proof* We first rewrite the objective in terms the total number of samples, $n$, across all state, action pairs, as opposed to the number of samples $m_{xa}$ for the $(X, A)$ pair:

$$\hat{Q}_{X,A} = -\min_{\alpha_c > 0} \left\{ \alpha_c \log \left( \frac{1}{m_{xa}} \sum_{R_j \in \mathcal{R}_{X,A}} \left[ \exp\left( \frac{-R_j}{\alpha_c} \right) \right] \right) + \alpha_c \delta_c \right\} \tag{26}$$

$$= -\min_{\alpha_c > 0} \left\{ \alpha_c \log \left( \frac{1}{p_0(x,a)n} \sum_{i=1}^n \left[ \mathbb{1}_{xa} \exp\left( \frac{-R_i}{\alpha_c} \right) \right] \right) + \alpha_c \delta_c \right\} \tag{27}$$

Where $\mathbb{1}_{xa}$ is the indicator function that is 1 if both $X_i = x$ and $A_i = a$, and 0 otherwise. We use the fact $\sum_{R_j \in \mathcal{R}_{X,A}} \left[ \exp\left( \frac{-R_j}{\alpha_c} \right) \right] = \sum_{i=1}^{n} \left[ \mathbb{1}_{xa} \exp\left( \frac{-R_i}{\alpha_c} \right) \right]$

Consider the value of $Q_{X,A}$ for a single $(X, A)$ pair. Consider the function $F_{xa}(\alpha_c) = \mathbb{E}[\mathbb{1}_{xa} \exp(C_i/\alpha_c)]$ with $f_{xa}(\alpha_c, C_i) = \mathbb{1}_{xa} \exp(C_i/\alpha_c)$. Then $f_{xa}(\alpha_c, C_i)$ is upper bounded by 1 and lower bounded by 0. Additionally note that $f_{xa}(\alpha_c, C_i)$ is locally L-Lipschitz with respect to $\alpha_c$ on $[\underline{\alpha}_c/2, 2\bar{\alpha}_c]$:

$$\frac{\partial f_{xa}(\alpha_c)}{\alpha_c} = -\frac{\mathbb{1}_{xa} C_i \exp(C_i/\alpha_c)}{\alpha_c^2} \tag{28}$$

$$\left| \frac{\partial f_{xa}(\alpha_c)}{\alpha_c} \right| \leq \frac{4 R_{max}}{\underline{\alpha}_c^2} \tag{29}$$

Additionally, from Lemma 5, we have $\bar{\alpha}_c = R_{max}/\delta_c$. Therefore, we can directly apply Lemma 4 to $F_{xa}(\alpha_c)$ and $\hat{F}_{xa} = \frac{1}{n} \sum_{i=1}^{n} f_{xa}(\alpha_c, C_i)$ with $d = 1$, $B = 2\bar{\alpha}_c = 2R_{max}/\delta_c$ and $L = 4R_{max}/\underline{\alpha}_c^2$ and conclude uniform convergence in $\alpha_c$ on the range $[\underline{\alpha}_c/2, 2\bar{\alpha}_c]$: with probability $1 - \delta'$:

$$\sup_{\alpha_c \in [\underline{\alpha}_c/2, 2\bar{\alpha}_c]} \left| F_{xa}(\alpha_c) - \hat{F}_{xa}(\alpha_c) \right| \leq \mathcal{O}\left( \frac{8(R_{max})^2}{\delta_c \underline{\alpha}_c^2} \sqrt{\frac{\log(n) \log(1/\delta')}{n}} \right) \tag{30}$$

Now define, $G_{xa}(\alpha_c) = \alpha_c \log\left( \frac{F_{xa}(\alpha_c)}{p_0(x,a)} \right) + \alpha_c \delta_c$. Note that $Q_{X,A} = \min_{\alpha_c \geq 0} G_{xa}(\alpha_c)$

$$\sup_{\alpha_c} \left| G_{xa}(\alpha_c) - \hat{G}_{xa}(\alpha_c) \right| = \sup_{\alpha_c} \left| \alpha_c (\log(F(\alpha_c)) - \log(\hat{F}(\alpha_c))) \right| \tag{31}$$

$$\leq 2\bar{\alpha}_c L \sup_{\alpha_c} \left| F(\alpha_c) - \hat{F}(\alpha_c) \right| \tag{32}$$

$$\leq \mathcal{O}\left( 2\bar{\alpha}_c \frac{8(R_{max})^2 L}{\delta_c \underline{\alpha}_c^2} \sqrt{\frac{\log(n) \log(\delta/2)}{n}} \right) \tag{33}$$

Where the second line utilizes the fact that log is locally L-Lipschitz and:

$$L = \frac{1}{\min\{\inf_{\alpha_c} \hat{F}(\alpha_c), \inf_{\alpha_c} F(\alpha_c)\}}$$

We now derive the quantity $L$. First note the expression $\exp(C_i/\alpha_c) \geq \exp(-R_{max}/\underline{\alpha}_c)$. With this we can write:

$$F_{xa}(\alpha_c) = p_0(x,a) \mathbb{E}[\exp(C_i/\alpha_c) | X_i = x, A_i = a] \tag{34}$$

$$\geq p_0(x,a) \exp(-R_{max}/\underline{\alpha}_c) \tag{35}$$

$$\text{and} \tag{36}$$

$$\hat{F}_{xa}(\alpha_c) = \frac{1}{n} \sum_{i=1}^{n} \mathbb{1}_{xa} \exp(C_i/\alpha_c) \tag{37}$$

$$\geq \exp(-R_{max}/\underline{\alpha}_c) \frac{1}{n} \sum_{i=1}^{n} \mathbb{1}_{xa} \tag{38}$$

$$= \exp(-R_{max}/\underline{\alpha}_c) \hat{p}_0(x,a) \tag{39}$$

Where $\hat{p}_0(x,a) = \frac{1}{n} \sum_{i=1}^{n} \mathbb{1}_{xa}$ is the empirical probability of observing $(X, A)$ in the dataset. Now we need to lower bound $\hat{p}_0(x,a)$. Let us lower bound the probability it is less than half of the true

value $p_0(x,a)/2$:

$$P\left(\hat{p}_0(x,a) \le \frac{p_0(x,a)}{2}\right) = P\left(-\hat{p}_0(x,a) \ge -\frac{p_0(x,a)}{2}\right) \tag{40}$$

$$= P\left(p_0(x,a) - \hat{p}_0(x,a) \ge p_0(x,a) - \frac{p_0(x,a)}{2}\right) \tag{41}$$

$$= P\left(p_0(x,a) - \hat{p}_0(x,a) \ge \frac{p_0(x,a)}{2}\right) \tag{42}$$

$$\overset{(1)}{\le} \exp\left(-2n\left(\frac{p_0(x,a)}{2}\right)^2\right) \tag{43}$$

$$= \exp\left(-\frac{np_0(x,a)^2}{2}\right) \tag{44}$$

Where step (1) applies Hoeffding inequality on the binary random variables $\mathbb{1}_{xa}$ and with $t = \frac{p_0(x,a)}{2}$

We can now calculate the minimum number of timesteps such that $P\left(\hat{p}_0(x,a) \le \frac{p_0(x,a)}{2}\right) \le \delta'$:

$$\exp\left(-\frac{p_0(x,a)^2}{2n}\right) \le \delta' \xrightarrow[\text{Solve for n}]{} n \ge \frac{2\log(1/\delta')}{p_0(x,a)^2} \tag{45}$$

Therefore for $n \ge \frac{2\log(1/\delta')}{p_0(x,a)^2}$, with probability $1 - \delta$ we have $\hat{p}_0(x,a) \ge \frac{p_0(x,a)}{2}$. We therefore have:

$$L = \frac{2}{\exp\left(\frac{-R_{max}}{\underline{\alpha}_c}\right)p_0(x,a)} \ge \frac{2}{\exp\left(\frac{-R_{max}}{\underline{\alpha}_c}\right)\underline{p}_0(x,a)} \tag{46}$$

We can now put everything together for the constant from equation 33:

$$c = 2\bar{\alpha}_c \frac{8(R_{max})^2 L}{\delta_c \underline{\alpha}_c^2} = \frac{32(R_{max})^3}{\delta_c^2 \underline{\alpha}_c^2 \exp\left(\frac{-R_{max}}{\underline{\alpha}_c}\right)\underline{p}_0(x,a)} \tag{47}$$

Let $\hat{\alpha}_c = \arg\min_{\alpha_c} \hat{G}_{xa}(\alpha_c)$. Note that $\hat{Q}_{X,A} = \hat{G}_{xa}(\hat{\alpha}_c)$. Let $\alpha_c^* = \arg\min_{\alpha_c} G_{xa}(\alpha_c)$. Note that $Q_{X,A} = G_{xa}(\alpha_c^*)$.

It then follows:

$$G_{xa}(\hat{\alpha}_c) - \hat{G}_{xa}(\hat{\alpha}_c) \le \mathcal{O}\left(c\sqrt{\frac{\log(n)\log(1/\delta')}{n}}\right) \tag{48}$$

$$G_{xa}(\hat{\alpha}_c) - \hat{G}_{xa}(\hat{\alpha}_c) + G_{xa}(\alpha_c^*) \le G_{xa}(\alpha_c^*) + \mathcal{O}\left(c\sqrt{\frac{\log(n)\log(1/\delta')}{n}}\right) \tag{49}$$

$$G_{xa}(\alpha_c^*) - \hat{G}_{xa}(\hat{\alpha}_c) \le G_{xa}(\alpha_c^*) - G_{xa}(\hat{\alpha}_c) + \mathcal{O}\left(c\sqrt{\frac{\log(n)\log(1/\delta')}{n}}\right) \tag{50}$$

$$Q_{X,A} - \hat{Q}_{X,A} \le \underbrace{G_{xa}(\alpha_c^*) - G_{xa}(\hat{\alpha}_c)}_{\le 0} + \mathcal{O}\left(c\sqrt{\frac{\log(n)\log(1/\delta')}{n}}\right) \tag{51}$$

$$Q_{X,A} - \hat{Q}_{X,A} \le \mathcal{O}\left(c\sqrt{\frac{\log(n)\log(1/\delta')}{n}}\right) \tag{52}$$

Similarly we can start with $G_{xa}(\alpha_c^*) - \hat{G}_{xa}(\alpha_c^*) \le \mathcal{O}\left(c\sqrt{\frac{\log(n)\log(1/\delta')}{n}}\right)$ and prove the other side $\hat{Q}_{X,A} - Q_{X,A} \le \mathcal{O}\left(c\sqrt{\frac{\log(n)\log(1/\delta')}{n}}\right)$.

By setting $\delta' = \delta/2$, and taking the union bound over the two sides, the desired lemma result follows. $\square$

**Note about $\underline{\alpha}_c$:** In this work we only consider cases where $\alpha_c > 0$. $\alpha_c$ is strictly bounded away from 0 by some $\underline{\alpha}_c > 0$. This occurs either when rewards have continuous support, or, in the case of discrete reward support, when $\delta_c$ is small enough such that the uncertainty set does not include the distribution with all mass at a single support value.

Since we are considering the case of discrete, finite context and action sets, we can now take a union bound over all $(X, A)$ pairs and achieve joint uniform convergence over all $(X, A)$ pairs to arrive at the following result:

**Lemma 7** (Uniform convergence over all $(X, A)$). *When $n \geq \frac{(2 \log(2|\mathcal{X}||\mathcal{A}|/\delta))}{(\underline{p}_0(x,a))^2}$, the following holds with probability of at least $1 - \delta$:*

$$\max_{(X,A)} \left| Q_{X,A} - \hat{Q}_{X,A} \right| \leq \mathcal{O}\left( c\sqrt{\frac{\log(n) \log(2|\mathcal{X}||\mathcal{A}|/\delta)}{n}} \right) \tag{53}$$

*Where*

$$c = \frac{32(R_{max})^3}{\delta_c^2 \underline{\alpha}_c^2 \exp\left(\frac{-R_{max}}{\underline{\alpha}_c}\right) \underline{p}_0(x,a)} \tag{54}$$

With this, we continue the proof, recall our goal is to optimize the distributionally robust objective:

$$\hat{V} = \min_{\pi, \alpha_x \geq 0} \left\{ \hat{J}(\pi, \alpha_x) \right\} \quad \text{and} \quad V(\pi) = \min_{\pi, \alpha_x \geq 0} \left\{ J(\pi, \alpha_x) \right\} \tag{55}$$

Where

$$\hat{J}(\pi, \alpha_x) = \alpha_x \log \mathbb{E}\left[ \exp\left( \frac{-\hat{Q}_{X,\pi}^{avg}}{\alpha_x} \right) \right] + \alpha_x \delta_x \tag{56}$$

$$J(\pi, \alpha_x) = \alpha_x \log \mathbb{E}\left[ \exp\left( \frac{-Q_{X,\pi}^{avg}}{\alpha_x} \right) \right] + \alpha_x \delta_x \tag{57}$$

And for a fixed policy $\pi$ and a context $X$

$$\hat{Q}_{X,\pi}^{avg} = \sum_a \pi(a|X)\hat{Q}_{X,a} \quad \text{and} \quad Q_{X,\pi}^{avg} = \sum_a \pi(a|X)Q_{X,a} \tag{58}$$

We will next show the following result which gives uniform convergence of $J(\pi, \alpha_x)$ in $\pi$ and $\alpha_x$.

**Lemma 8** (Uniform convergence of optimization objective). *With probability of at least $1 - \delta$ and with $n \geq \frac{(2 \log(2|\mathcal{X}||\mathcal{A}|/\delta))}{(\underline{p}_0(x,a))^2}$:*

$$\sup_{\pi, \alpha_x} \left| \hat{J}(\pi, \alpha_x) - J(\pi, \alpha_x) \right| \leq \mathcal{O}\left( c\sqrt{\frac{\log(n) \log\left( \frac{2|\mathcal{X}||\mathcal{A}|}{\delta} \right)}{n}} \right) \tag{59}$$

*Proof:* For all steps, assume $n \geq \frac{(2 \log(2|\mathcal{X}||\mathcal{A}|/\delta))}{(\underline{p}_0(x,a))^2}$. Because $\sum_a \pi(a|X) = 1$, $Q_{X,\pi}^{avg}$ is a weighted average of $Q_{X,A}$.

$$\left| \hat{Q}_{X,\pi}^{avg} - Q_{X,\pi}^{avg} \right| \leq \max \left| Q_{X,A} - \hat{Q}_{X,A} \right| \leq \mathcal{O}\left( c\sqrt{\frac{\log(n) \log\left( \frac{2|\mathcal{X}||\mathcal{A}|}{\delta} \right)}{n}} \right) \tag{60}$$

Additionally because Lemma 7 holds for simultaneously all $(X, A)$ pairs, and every $\hat{Q}_{i,\pi}^{avg}$ depends on the same set of $Q_{X,A}$, we have the above result holds with probability with $1 - \delta$ for all $i$. Formally, with probability $1 - \delta$:

$$\max_X \left| \hat{Q}_{X,\pi}^{avg} - Q_{X,\pi}^{avg} \right| \leq \max \left| Q_{X,A} - \hat{Q}_{X,A} \right| \leq \mathcal{O}\left( c\sqrt{\frac{\log(n)\log\left(\frac{2|\mathcal{X}||\mathcal{A}|}{\delta}\right)}{n}} \right) \tag{61}$$

Then we have:

$$\sup \left| \hat{J}(\pi, \alpha_x) - J(\pi, \alpha_x) \right| = \alpha_x \sup \left| \log \mathbb{E}\left[ \exp\left( \frac{-\hat{Q}_{X,\pi}^{avg}}{\alpha_x} \right) \right] - \log \mathbb{E}\left[ \exp\left( \frac{-Q_{X,\pi}^{avg}}{\alpha_x} \right) \right] \right| \tag{62}$$

$$= \alpha_x \sup \left| \left( \log \mathbb{E}\left[ \frac{\exp\left( \frac{-\hat{Q}_{i,\pi}^{avg}}{\alpha_x} \right)}{\exp\left( \frac{-Q_{i,\pi}^{avg}}{\alpha_x} \right)} \right] \right) \right| \tag{63}$$

$$= \alpha_x \sup \left| \left( \log \mathbb{E}\left[ \exp\left( \frac{Q_{i,\pi}^{avg} - \hat{Q}_{i,\pi}^{avg}}{\alpha_x} \right) \right] \right) \right| \tag{64}$$

$$\leq \alpha_x \left( \log \mathbb{E}\left[ \exp\left( \frac{\sup_i \left| Q_{i,\pi}^{avg} - \hat{Q}_{i,\pi}^{avg} \right|}{\alpha_x} \right) \right] \right) \tag{65}$$

$$\leq \alpha_x \left( \log \left( \exp\left( \frac{\mathcal{O}\left( c\sqrt{\frac{\log(n)\log\left(\frac{2|\mathcal{X}||\mathcal{A}|}{\delta}\right)}{n}} \right)}{\alpha_x} \right) \right) \right) \tag{66}$$

$$= \mathcal{O}\left( c\sqrt{\frac{\log(n)\log\left(\frac{2|\mathcal{X}||\mathcal{A}|}{\delta}\right)}{n}} \right) \tag{67}$$

Which is the desired result. $\qquad\square$

We can now show convergence in policy evaluation. We first rewrite the objectives, $\hat{V}(\pi)$ and $\hat{V}(\pi)$, using $J(\pi, \alpha_x)$ and $\hat{J}(\pi, \alpha_x)$. Given a fixed policy $\pi$:

$$\hat{V}(\pi) = \min_{\alpha_x \geq 0} \left\{ \hat{J}(\pi, \alpha_x) \right\} \quad \text{and} \quad V(\pi) = \min_{\alpha_x \geq 0} \left\{ J(\pi, \alpha_x) \right\} \tag{68}$$

We now restate the theorem:

**Theorem 2** (Convergence of policy evaluation). *For $n \geq \frac{(2\log(2|\mathcal{X}||\mathcal{A}|/\delta)}{(\underline{p}_0(x,a))^2}$, the following holds for any $\pi$ with probability of at least $1 - \delta$:*

$$\left| V(\pi) - \hat{V}(\pi) \right| \leq \mathcal{O}\left( c\sqrt{\frac{\log(n)\log\left(\frac{2|\mathcal{X}||\mathcal{A}|}{\delta}\right)}{n}} \right) \quad \text{where} \ \ c = \frac{32(R_{max})^3}{\delta_c^2 \underline{\alpha}_c^2 \exp(\frac{-R_{max}}{\underline{\alpha}_c})\underline{p}_0(x,a)} \tag{8}$$

*and $\underline{p}_0(x,a) = \min_{(x,a)} p_0(x,a) \geq \epsilon_\pi \epsilon_X$ is the minimum over probability of occurrence of (X,A) pairs, and $\underline{\alpha}_c$ is the minimum value for $\alpha_c$.*

*Proof* For all steps, assume $n \geq \frac{(2\log(2|\mathcal{X}||\mathcal{A}|/\delta)}{(\underline{p}_0(x,a))^2}$. Define:

$$\hat{\alpha}_x = \operatorname*{argmin}_{\alpha_x \geq 0} \left\{ \hat{J}(\pi, \alpha_x) \right\} \quad \text{and} \quad \alpha_x^* = \operatorname*{argmin}_{\alpha_x \geq 0} \left\{ J(\pi, \alpha_x) \right\} \tag{69}$$

It then follows:

$$J(\pi, \hat{\alpha}_x) - \hat{J}(\pi, \hat{\alpha}_x) \leq \mathcal{O}\left(c\sqrt{\frac{\log(n)\log\left(\frac{2|\mathcal{X}||\mathcal{A}|}{\delta}\right)}{n}}\right) \tag{70}$$

$$J(\pi, \hat{\alpha}_x) - \hat{J}(\pi, \hat{\alpha}_x) + J(\pi, \alpha_x^*) \leq J(\pi, \alpha_x^*) + \mathcal{O}\left(c\sqrt{\frac{\log(n)\log\left(\frac{2|\mathcal{X}||\mathcal{A}|}{\delta}\right)}{n}}\right) \tag{71}$$

$$J(\pi, \alpha_x^*) - \hat{J}(\pi, \hat{\alpha}_x) \leq J(\pi, \alpha_x^*) - J(\pi, \hat{\alpha}_x) + \mathcal{O}\left(c\sqrt{\frac{\log(n)\log\left(\frac{2|\mathcal{X}||\mathcal{A}|}{\delta}\right)}{n}}\right) \tag{72}$$

$$V(\pi) - \hat{V}(\pi) \leq \underbrace{J(\pi, \alpha_x^*) - J(\pi, \hat{\alpha}_x)}_{\leq 0 \text{ by definition of } \alpha_x^*} + \mathcal{O}\left(c\sqrt{\frac{\log(n)\log\left(\frac{2|\mathcal{X}||\mathcal{A}|}{\delta}\right)}{n}}\right) \tag{73}$$

$$V(\pi) - \hat{V}(\pi) \leq \mathcal{O}\left(c\sqrt{\frac{\log(n)\log\left(\frac{2|\mathcal{X}||\mathcal{A}|}{\delta}\right)}{n}}\right) \tag{74}$$

We can also show the other side:

$$\hat{J}(\pi, \alpha*_x) - J(\pi, \alpha_x^*) \leq \mathcal{O}\left(c\sqrt{\frac{\log(n)\log\left(\frac{2|\mathcal{X}||\mathcal{A}|}{\delta}\right)}{n}}\right) \tag{75}$$

$$\hat{J}(\pi, \alpha*_x) - J(\pi, \alpha_x^*) + \hat{J}(\pi, \hat{\alpha}_x) \leq \hat{J}(\pi, \hat{\alpha}_x) + \mathcal{O}\left(c\sqrt{\frac{\log(n)\log\left(\frac{2|\mathcal{X}||\mathcal{A}|}{\delta}\right)}{n}}\right) \tag{76}$$

$$\hat{J}(\pi, \hat{\alpha}_x) - J(\pi, \alpha_x^*) \leq \hat{J}(\pi, \hat{\alpha}_x) - \hat{J}(\pi, \alpha*_x) + \mathcal{O}\left(c\sqrt{\frac{\log(n)\log\left(\frac{2|\mathcal{X}||\mathcal{A}|}{\delta}\right)}{n}}\right) \tag{77}$$

$$\hat{V}(\pi) - V(\pi) \leq \underbrace{\hat{J}(\pi, \hat{\alpha}_x) - \hat{J}(\pi, \alpha*_x)}_{\leq 0 \text{ by definition of } \hat{\alpha}_x} + \mathcal{O}\left(c\sqrt{\frac{\log(n)\log\left(\frac{2|\mathcal{X}||\mathcal{A}|}{\delta}\right)}{n}}\right) \tag{78}$$

$$\hat{V}(\pi) - V(\pi) \leq \mathcal{O}\left(c\sqrt{\frac{\log(n)\log\left(\frac{2|\mathcal{X}||\mathcal{A}|}{\delta}\right)}{n}}\right) \tag{79}$$

Combining these two shows the result gives the desired result. $\qquad\square$

Now lastly consider the case of policy learning. Define

$$V^* = \min_{\pi, \alpha_x} J(\pi, \alpha_x) \quad \text{and} \quad \hat{V}^* = \min_{\pi, \alpha_x} \hat{J}(\pi, \alpha_x) \tag{80}$$

Additionally define optimal parameters:

$$(\pi^*, \alpha_x^*) = \operatorname*{argmin}_{\pi, \alpha_x} J(\pi, \alpha_x) \quad \text{and} \quad (\hat{\pi}, \hat{\alpha}_x) = \operatorname*{argmin}_{\pi, \alpha_x} \hat{J}(\pi, \alpha_x) \tag{81}$$

Finally, recall the the set of $\epsilon$-optimal policies $\Pi_\epsilon^* = \{\pi : V(\pi) \geq V^* - \epsilon\}$.

We first restate the theorem:

**Theorem 3** (Convergence of Policy Learning). *For $n \geq \frac{(2\log(2|\mathcal{X}||\mathcal{A}|/\delta)}{(\underline{p}_0(x,a))^2}$, the following holds with probability of at least $1 - \delta$:*

$$\left| \hat{V}^* - V^* \right| \leq \mathcal{O}\left( c\sqrt{\frac{\log(n)\log\left(\frac{2|\mathcal{X}||\mathcal{A}|}{\delta}\right)}{n}} \right) \tag{9}$$

*Equivalently this says*

$$P(\hat{\pi} \in \Pi_\epsilon^*) > 1 - \mathcal{O}\left( 2|\mathcal{X}||\mathcal{A}| \exp\left( -\frac{n\epsilon^2}{\log(n)c^2} \right) \right) \tag{10}$$

*Proof*: This proof is similar to the proof of policy evaluation of the previous theorem. The first side of the inequality

$$J(\hat{\pi}, \hat{\alpha}_x) - \hat{J}(\hat{\pi}, \hat{\alpha}_x) \leq \mathcal{O}\left( c\sqrt{\frac{\log(n)\log\left(\frac{2|\mathcal{X}||\mathcal{A}|}{\delta}\right)}{n}} \right) \tag{82}$$

$$J(\hat{\pi}, \hat{\alpha}_x) - \hat{J}(\hat{\pi}, \hat{\alpha}_x) + J(\pi^*, \alpha_x^*) \leq J(\pi *' \alpha_x^*) + \mathcal{O}\left( c\sqrt{\frac{\log(n)\log\left(\frac{2|\mathcal{X}||\mathcal{A}|}{\delta}\right)}{n}} \right) \tag{83}$$

$$J(\pi^*, \alpha_x^*) - \hat{J}(\hat{\pi}, \hat{\alpha}_x) \leq J(\pi^*, \alpha_x^*) - \hat{J}(\hat{\pi}, \hat{\alpha}_x) + \mathcal{O}\left( c\sqrt{\frac{\log(n)\log\left(\frac{2|\mathcal{X}||\mathcal{A}|}{\delta}\right)}{n}} \right) \tag{84}$$

$$V^* - \hat{V}^* \leq \underbrace{J(\pi^*, \alpha_x^*) - J(\hat{\pi}, \hat{\alpha}_x)}_{\leq 0 \text{ by definition of } (\pi^*, \alpha_x^*)} + \mathcal{O}\left( c\sqrt{\frac{\log(n)\log\left(\frac{2|\mathcal{X}||\mathcal{A}|}{\delta}\right)}{n}} \right) \tag{85}$$

$$V^* - \hat{V}^* \leq \mathcal{O}\left( c\sqrt{\frac{\log(n)\log\left(\frac{2|\mathcal{X}||\mathcal{A}|}{\delta}\right)}{n}} \right) \tag{86}$$

The other side of the inequality:

$$\hat{J}(\pi^*, \alpha*_x) - J(\pi^*, \alpha_x^*) \le \mathcal{O}\left( c\sqrt{\frac{\log(n) \log\left(\frac{2|\mathcal{X}||\mathcal{A}|}{\delta}\right)}{n}} \right) \tag{87}$$

$$\hat{J}(\pi^*, \alpha*_x) - J(\pi^*, \alpha_x^*) + \hat{J}(\hat{\pi}, \hat{\alpha}_x) \le \hat{J}(\hat{\pi}, \hat{\alpha}_x) + \mathcal{O}\left( c\sqrt{\frac{\log(n) \log\left(\frac{2|\mathcal{X}||\mathcal{A}|}{\delta}\right)}{n}} \right) \tag{88}$$

$$\hat{J}(\hat{\pi}, \hat{\alpha}_x) - J(\pi^*, \alpha_x^*) \le \hat{J}(\hat{\pi}, \hat{\alpha}_x) - \hat{J}(\pi^*, \alpha*_x) + \mathcal{O}\left( c\sqrt{\frac{\log(n) \log\left(\frac{2|\mathcal{X}||\mathcal{A}|}{\delta}\right)}{n}} \right) \tag{89}$$

$$\hat{V}^* - V^* \le \underbrace{\hat{J}(\hat{\pi}, \hat{\alpha}_x) - \hat{J}(\pi^*, \alpha*_x)}_{\le 0 \text{ by definition of } (\hat{\pi}, \hat{\alpha}_x)} + \mathcal{O}\left( c\sqrt{\frac{\log(n) \log\left(\frac{2|\mathcal{X}||\mathcal{A}|}{\delta}\right)}{n}} \right) \tag{90}$$

$$\hat{V}^* - V^* \le \mathcal{O}\left( c\sqrt{\frac{\log(n) \log\left(\frac{2|\mathcal{X}||\mathcal{A}|}{\delta}\right)}{n}} \right) \tag{91}$$

Combining these two shows the result gives the desired result. $\qquad\square$