# OpenReview forum: "Factored DRO: Factored Distributionally Robust Policies for Contextual Bandits"
_NeurIPS.cc/2022/Conference — NeurIPS 2022 Accept_

### Official Review · Reviewer_tRpt · 2022-07-10

**Rating:** 6
**Confidence:** 3
**Soundness:** 3 good
**Presentation:** 3 good
**Contribution:** 3 good

**Summary:**

This paper proposes an algorithm to learn a contextual bandit policy in the batch setting when there is a distributional shift between training and deployment. The main motivation of the proposed approach is the benefit of constraining the factors of the distribution of the context and the reward.

Based on the existing theoretical work on the minimax approach to DRO, the paper proposes a procedure for policy evaluation and improvement with convergence analysis. With other estimation details filled, FDRO is empirically evaluated against other baselines highlighting the strength of the proposed approach.

**Questions:**

- It seems that MGF approximation seems working well in practice. However, (considering a bit of theoretical flavor of this paper) in theory, MGF is not always guaranteed to exist. Then if the nominal (true) distribution has the MGF, could it be possible to say that a distribution of $\epsilon$ KL-ball of the nominal distribution had MFG?
- For continuous contexts, binning approach has been proposed. I was wondering if there is any other approach the author tried, for example, k-NN like smoothing, estimating reward distribution for a given context using (R,C,A) triplets from nearby contexts.
- When there is a joint shift in the context and the reward, how can FDRO be compared with the joint constraining, e.g. baseline-DRO?
- Does theorem 2 hold uniformly for $\pi$? Since the uniform convergence is mentioned on line 208, it would be better to clarify this.
- Factor constraining is basically considering KL-ball for each factor from some factorization of the original full distribution. In the full KL-ball of the same factorization, it is KL-distance on context distribution and expected KL-distance on context-action conditioned reward distribution. Then this can motivate a variation of FDRO, in which not setting the uniform (minimax) bound of KL-distance on context-action conditioned reward distribution, expected (Bayesian) bound of KL-distance on reward distribution. How would this be compared with the FDRO, any thought on this?
- In contrast to context shift, reward shift $\delta_c$ is fixed to $0.001$. How is the sensitivity to this hyperparameter?

**Limitations:**

The main motivation seems that the training tractability of factored constraining over joint constraining. However, the benefit of the current approach does not seem to be easily translated into continuous contexts making the estimation of reward distribution difficult. Still, the benefit of FDRO in discrete contexts is noticeable. Maybe its extension of the currently proposed binning approach for continuous contexts can be future work. In relation to this limitation, I am curious about the extent of the FDRO in discrete reward settings, for example, in the case where there are a larger number of discrete contexts and discrete actions, how FDRO would perform. Rather than asking to perform additional experiments for this, I would be happy if the authors can share their thought and previous analyses on this.

**Strengths And Weaknesses:**

Strengths

- The proposed policy iteration is well-motivated by a previously established theoretical work.
- Also, the convergence analysis of the proposed method is provided.
- Empirically it is demonstrated that the factor constraining generally does not harm the policy iteration compared to the joint constraining except for a slim edge case (the same train/test).

Weaknesses

- Rather than calling them weakness, I put relevant questions in below box.
- There are some typos
  * in eq.(6) $\frac{-R}{\alpha_c}$ subscript $j$ of $R$ missing
  * at line 290, $V^* = \min_{\pi} V(\pi)$ which seems to have to be $\max_{\pi}$,
  * in line 102 'We assume we the',
  * in eq.(4), closing curly/square parenthesis are swapped, etc.

---

> ### Author Response · Authors · 2022-07-28
> **Quick question: references for "Factor Constraining"**
>
> Thank you so much for your thoughtful feedback! We are currently working on addressing your comments. To ensure we address the method you were thinking of, we were wondering if you would be able to provide a few references to the "Factor constraining" method mentioned.

---

> ### Author Response · Authors · 2022-08-02
> **Response to Reviewer tRpt**
>
> Thank you for your thorough and helpful feedback!
> - **Sensitivity to $\delta_c$** Thank you for the nice suggestion! We update Figure 1f and the Appendix (Figure 3a) to include plots showing the performance of the Scene and simulated Bernoulli environments for different values of the reward shift radius hyperparameter, $\delta_c$ (0.001, 0.1, 0.25). Empirically we found performance to not vary too much based off of choice of $\delta_c$.
> - **Other Smoothing Approaches** Thank you for the suggestion. While we focus on evaluating with the Taylor expansion approximation for smoothing in continuous contexts, we agree that other approaches such as k-NN smoothing would most likely work. We did not try this approach, but we expect it can have good performance and it would be great to consider these methods in future work.
> - **The existence of the MGF** Thank you for your question. We first quickly note that we do not need the data distributions to have a MGF. Instead we notice a component of our estimator, $E[\exp(-R)/\alpha_c]$ naturally defines a MGF over the variable $c_{xa} \sim P(-R|X,A)$. We then estimate this value with a Taylor series approximation. We realize that using the term MGF may be confusing and we have changed the wording in section 5 to clarify this.  The key question is then the validity of the Taylor expansion of $E[\exp(t c_{xa})$ where $t = 1/\alpha_c$. Because we assume rewards $R$ are bounded, it follows $c_{xa}$ is subgaussian so the MGF exists and is finite, and there is a valid Taylor expansion for $\alpha_c > 0$. Let us know if this answers your question.
> - **Theory** Thank you for the question! We updated our theoretical analysis and show non-asymptotic convergence. We have updated Theorem 2 and 3 to include rates of convergence and we show our algorithm has $\sqrt{\frac{log(n)}{n}}$ dependence on n.
> - **Joint Shift**  Thank you for bringing up this point. We update our discussions in Section 6.2 to provide more discussion comparing baseline-DRO and FDRO as well as the equivalence of solutions (under the heading 'Discussion: When to use FDRO vs baseline-DRO (equivalence of solutions)). To summarize: the KL-divergence chain rule decomposition: $D_{kl}(p_0(R, X)|| p(R,X)) = D_{kl}(p_0(X)||p(X)) + E_{p_0(x)} [D_{kl}(p_0(R|X)||p(R|X))]$. Recall $D_{kl}(p_0(R, X)|| p(R,X)) = \delta$, $D_{kl}(p_0(X)||p(X)) = \delta_x$, $D_{kl}(p_0(R|X)||p(R|X)) = \delta_c$. We can see, when $\delta_c$ is the same for each context, we have $\delta = \delta_x + \delta_c$. When $\delta$ of baseline-DRO is set to $\delta_x + \delta_c$ baseline-DRO and FDRO consider distributions the same KL distance away, with FDRO placing additional constraints on the set of distributions searched. From the decomposition, we can see, with additional stipulations, we can set a family of $\delta_x$ and $\delta_c$ values for FDRO to achieve the same solution of baseline-DRO for each $\delta$ (we give details in appendix). However, for a given $\delta_x$ and $\delta_c$ it is not possible to find a $\delta$ for baseline-DRO that achieves the same solution. However we do expect FRDO to be particularly useful when there are independent shifts in the reward and context distributions, which is a common setting, such as shifting locations, as the $\delta_x$ and $\delta_c$ are easier to set.
> - **Continuous/large contexts spaces:** Thank you for bringing up this point. We first want to clarify that the practical method proposed in Section 5 works for continuous contexts. All the empirical evaluations utilize this method and are in continuous context settings and we use the practical method for FDRO results.   To answer your question about larger context dimensions and action spaces, we expect our method to work well in larger context spaces, given enough data. We note some of the datasets we consider have large feature spaces such as the Scene dataset where the context space has 294 continuous features. Due to the small dataset size, we use random projection to project this down to 3 continuous features. To illustrate our method's ability to handle larger feature dimensions, we update our appendix to include a figure (Figure 2c) where we project to 10 continuous features instead (we expect our method to work for larger dimensions as well, but found it hard to increase dimension past this due to the small amount of data).
>
> **Did the above answer your questions? We also welcome additional questions or feedback.**

---

> > ### Comment · Reviewer_tRpt · 2022-08-08
> > **Thanks for the detailed answers to my questions**
> >
> > The answers address most of my concerns. I do not have further concerns. Even though this is not the field I am thoroughly covering, I am still inclined to the acceptance of this paper. Such factorization seems a natural choice if there is no strong evidence for the need for modeling complex non-factored relations. Supporting theory seems without serious errors. Although I can see some concerns about the empirical evaluations from another reviewer, I will keep my positive score.

---

### Official Review · Reviewer_yHTP · 2022-07-12

**Rating:** 6
**Confidence:** 3
**Soundness:** 3 good
**Presentation:** 3 good
**Contribution:** 2 fair

**Summary:**

This paper provides a method (Factored-DRO) that learns a distributionally robust policy over separately handled shifts in context and reward distributions.

**Questions:**

- Why is it the baseline-DRO can be overly conservative compared to FDRO? Are there situations where FDRO can also be overly conservative (or overly optimistic)?

- Is FDRO guaranteed to be always better than baseline-DRO in test, and why? If not, is there a situation where we can expect baseline-DRO to do better?

- Is it possible to set $\delta$ and $\delta_x, \delta_c$ such that baseline-DRO and FDRO will have equivalent performance?

- As the authors mention, one limitation common to DRO methods is setting the $\delta$ parameter. Since FDRO has two parameters to set, is it more sensitive to improperly set values? E.g. if $\delta_x$ and $\delta_c$ in FDRO are set too large, does the error compound, leading to worse performance, compared to baseline-DRO when $\delta$ is similarly set too large?

- In this light, does this imply that FDRO may be less advantageous to use for users who have less information about the distribution shifts in their application of interest? Should the user be more or less conservative in setting the parameter $\delta$ in FDRO vs baseline-DRO?

- Why are finite sample guarantees given for reward shift but asymptotic for context shift? Is it possible to obtain finite sample guarantees for policy evaluation and learning (Theorems 2 and 3) versus convergence of reward shift estimates, and if not, what is the barrier?

**Limitations:**

Aside from the points addressed in my comments above, the authors have adequately addressed the limitations and impacts of their work.

**Strengths And Weaknesses:**

**Strengths**

Overall, I find the motivations to be convincing, the methodology to be reasonable, and the paper to be generally well-written.


**Weaknesses**

I would be interested in a more in-depth conceptual, theoretical, and practical comparison between FDRO and its closest point of comparison baseline-DRO (Si et al). For example, while FDRO has the advantage of being able to separate out shifts in context and reward distributions, by requiring more parameters to be set, is it more sensitive to improperly set parameters, i.e. does the error compound? Does this imply that FDRO may be less advantageous to use for users who have less information about the distribution shifts in their application of interest?

Please see my questions below.

---

> ### Author Response · Authors · 2022-08-02
> **Response to Reviewer yHTP**
>
> Thank you for bringing up these important points and questions. To address your questions about comparisons under our algorithm and baseline-DRO, we added an additional discussion, "Discussion: When to use FDRO vs baseline-DRO (equivalence of solutions)" to Section 6.2. We answer your questions (potentially out of order) below.
>
> - **Setting parameters for equivalent performance** Thank you for this question! In general, it is not always possible, for a given $\delta_c$ and $\delta_x$, to set $\delta$ (in baseline DRO) such that the resulting algorithm will output the same policy as our method. Consider the KL-divergence chain rule decomposition: $D_{kl}(p_0(R, X)|| p(R,X)) = D_{kl}(p_0(X)||p(X)) + E_{p_0(x)} [D_{kl}(p_0(R|X)||p(R|X))]$. Recall $D_{kl}(p_0(R, X)|| p(R,X)) = \delta$, $D_{kl}(p_0(X)||p(X)) = \delta_x$, $D_{kl}(p_0(R|X)||p(R|X)) = \delta_c$. We can see, when $\delta_c$ is the same for each context, we have $\delta = \delta_x + \delta_c$. When $\delta$ of baseline-DRO is set to $\delta_x + \delta_c$ baseline-DRO and FDRO consider distributions at most the same KL distance away, with FDRO placing additional constraints on the set of distributions searched. From the decomposition, we can see, with additional stipulations effort, one can set a family of $\delta_x$ and $\delta_c$ values for FDRO to achieve the same solution of baseline-DRO for each $\delta$ (we give additional details in appendix). However, for a given $\delta_x$ and $\delta_c$ it is not possible to find a $\delta$ for baseline-DRO that achieves the same solution. We additionally update our appendix (supplementary material) to provide a detailed example in a simple 2 context, discrete reward setting in section A.4.4 that examines the difference of behavior under the two formulations as well as the conditions for equivalence.
>
> - **Why is baseline-DRO overly conservative? When is FDRO worse?**:  From the decomposition given previously, we see Baseline-DRO has less constraints and can, for example, place the shift “budget” ($\delta_x + \delta_c$) all in the rewards or all in the contexts if that is the worst-case. This causes the degenerate performance in the binary rewards case for baseline-DRO mentioned in the text where reward shift dominates. FDRO can set limits on how much reward shift is considered and ensure context shift is considered, allowing us to overcome this. That said, as you mentioned FDRO does have more parameters which makes it more difficult to set. There may indeed be cases where FDRO's parameters are set incorrectly and perform worse than baseline-DRO.
>
> - **When to use FDRO vs baseline-DRO:** The degenerate binary example above informs us that generally, if the algorithm designer would like to ensure a certain amount of reward and context shift are considered (for example, as we need to in our updated Voting experiment with binary rewards, where we want to use data from one city to inform policies for other cities), then it may be beneficial to use FDRO and invest the additional effort to decompose $\delta$ between $\delta_x$ and $\delta_c$. Additionally, we expect FRDO to be particularly useful when there are independent shifts in the reward and context distributions, which is a common setting, such as shifting locations, as there, $\delta_x$ and $\delta_c$ are easier to set.
>
> - **Error in FDRO and baseline-DRO:**  As previously mentioned, when we set baseline-DRO's $\delta = \delta_x + \delta_c$, then both FDRO and baseline-DRO arrive at solutions that search over worst-case distributions at most a distance of $\delta$ away from the data generation distribution. They arrive at different solutions, as FDRO searches a more constrained space, however it is difficult without knowing the intended deployment distributions ahead of time to know which solution will work better.
>
> - **Sensitivity of parameters of FDRO:** Thank you for bringing up this point. We update Figure 1f and the Appendix to include plots showing the performance of the Scene and simulated Bernoulli environments for different values of the reward shift radius hyperparameter, $\delta_c$ (0.001, 0.1, 0.25). Empirically we found performance to not vary too much based off of choice of $\delta_c$.
>
> - **Theoretical Analysis:** Thank you for highlighting the need for updated theoretical analysis. We updated our theoretical analysis and we have updated Theorem 2 and 3 to include rates of convergence that depend on the number of samples n. Our analysis is non-asymptotic and we show our algorithm has $\sqrt{\frac{log(n)}{n}}$ dependence on n.
>
> **Did the above answer your questions? We also welcome additional questions or feedback.**

---

### Official Review · Reviewer_R2tu · 2022-07-18

**Rating:** 6
**Confidence:** 3
**Soundness:** 3 good
**Presentation:** 3 good
**Contribution:** 3 good

**Summary:**

This paper proposes an improvement of distributionally robust offline learning for contextual Bandit problem. The authors formulates the robust learning problem by proposing a nested robustness criteria involving context drfit and drift in reward distribution. It builds upon the recent work but expands to the 'factored' form to make the robustness specification more fine-grained.

**Questions:**

Can you concretely show how your algorithm can fix the issue highlighted in line159-161?

**Limitations:**

I think the authors have done a good job in listing the limitations and I have nothing further to add.

**Strengths And Weaknesses:**

To me this is a good piece of work that builds upon the existing work and make it more fine-grained. The motivation of the problem has been clearly articulated and the mathematical super-structure has been nicely laid out thorugh both the main text and the appendix. This is not in my field of expertise, but I think the amount of novelty and the rigor is enough for an acceptance.

At some points it feels like a jump, especially from eq 2 to eq 4 and 5. It would be nice to provide more detailed steps.

---

> ### Author Response · Authors · 2022-08-02
> **Response to Reviewer R2tu**
>
> Thank you for your helpful feedback! We added additional prose between equation 2 and equations 4 and 5.
>
> - **How FDRO overcomes the degenerate case** Thank you for asking this question! To address this question, we update our discussions in Section 6.2 to provide more discussion comparing baseline-DRO and FDRO as well as the equivalence of solutions (under the heading 'Discussion: When to use FDRO vs baseline-DRO (equivalence of solutions)).  In summary, we can see when we set the baseline-DRO parameter to $\delta = \delta_x + \delta_c$ baseline-DRO and FDRO consider distributions the same KL distance away, with FDRO placing additional constraints on the set of distributions searched. Baseline-DRO has less constraints and can, for example, place the shift “budget” ($\delta_x + \delta_c$) all in the rewards or all in the contexts if that is the worst case. This causes the degenerate performance in the binary rewards case for baseline-DRO where reward shift dominates. FDRO can set limits on how much reward shift is considered and ensures context shift is considered, allowing FDRO to overcome this.
>
> **Did the above answer your questions? We also welcome additional questions.**

---

> > ### Comment · Reviewer_R2tu · 2022-08-08
> > **Thank you**
> >
> > Thanks for the clarification. I do not have further questions.

---

### Official Review · Reviewer_WGag · 2022-07-19

**Rating:** 5
**Confidence:** 3
**Soundness:** 3 good
**Presentation:** 2 fair
**Contribution:** 3 good

**Summary:**

This work considers the contextual bandit problem with shifts on the distributions of both contexts and reward generators. An analysis on asymptotical optimality is provided. A practical algorithm as well as the empirical evaluations are presented.

**Questions:**

Sometimes the distribution shift on the context does not matter under some weak assumptions, e.g., LinUCB works on linear contextual bandit with any context distribution. Therefore, a relevant question is when we should take context distribution shift into consideration?

**Limitations:**

no obvious limitation.

**Strengths And Weaknesses:**

Strength:
This work is well-motivated. The problem of distribution shift is significant in practice.
The theoretical result is satisfying, even though it analyses the asymptotic behavior of the algorithm.
This work attempts to provide practical version of the algorithm.

Weakness:
The proposed algorithm seems not very practical as it relies on Taylor expansion, especially when the underlying functions have many parameters.

---

> ### Author Response · Authors · 2022-08-02
> **Response to Reviewer WGag**
>
> Thank you for your helpful comments!
> - **Taylor expansion impracticality**: Thank you for bringing up this concern! In our method, we use a Taylor expansion approximation of a finite order m. We note that while our context may be high dimensional, we are actually only taking the Taylor expansion with respect to the scalar $\alpha_c$. Therefore the difficulty of the Taylor approximation does not scale with the context dimension space. We also wanted to clarify something that might have been confusing in our submitted text: we mention that one could train a model to predict each moment which indeed can be very computationally burdensome. However many common models can predict multidimensional outputs (ex. kernel regression, neural networks) and for those methods, we actually only need to train a single model that predicts all the moments. This is much more computationally efficient and for all our experiments we take this approach. We have updated Section 5 to reflect these points.
> - **When to consider context shift?**: Thank you for the opportunity to clarify this. We agree with the reviewer that there are cases where distribution shift indeed may not matter (e.g. see our section 3  “Note about misspecification”).  Distribution shift is important when there is misspecification and the underlying policy class does not perfectly capture the underlying process. In these cases, the learned model encodes the biases of the population distribution. This can occur both when the function class is underspecified or when the function class is overspecified/when there is not enough data so regularization is needed. You are correct that when both the underlying policy function class can perfectly capture the underlying process and there is sufficient data such that regularization is not needed, then we do not need to consider shifts as we can perfectly choose the best action for each context. However many real-world processes (for example human behavior) are often complex and hard/impractical to model so often models may be misspecified. For example, in our empirical evaluation (Figure 1) we compare to a common baseline in the batch context setting that does not consider shifts (baseline-IS). In the real-world shift between cities for the Voting dataset experiment (updated Figure 1c), there is a significant shift that our method does better on.
> - **Updated Theoretical Analysis**: We updated our theoretical analysis and we have updated Theorem 2 and 3 to include rates of convergence which depend on the number of samples n. Our analysis is non-asymptotic and we show our algorithm has $\sqrt{\frac{log(n)}{n}}$ dependence on n.
>
> **Did the above answer your questions? We also welcome additional questions.**

---

### Official Review · Reviewer_kRMP · 2022-07-22

**Rating:** 6
**Confidence:** 3
**Soundness:** 3 good
**Presentation:** 4 excellent
**Contribution:** 3 good

**Summary:**

The paper studies the problem of distributionally robust batch policy optimization in contextual bandits, where the distributional shift occurs over both the context and reward distribution. Under bounded error in KL-divergence, the authors show that the value of a policy can be optimized by considering the dual objective under both distribution shifts. The authors deviate from prior work by considering the two distribution shifts separately in a factored optimization objective.

The authors theoretically show that that the estimated value of a policy converges to the true one asymptotically, and empirically demonstrate its effective against existing baselines in several real-world domains; in doing so, the authors propose how their proposed objective can be estimated outside of tabular domains (i.e. when context is a continuous vector).

**Questions:**

I was wondering if sample complexity bounds can be derived for Theorem 2 and 3 akin to Lemma 1? Specifically, is there something that prevents the authors from applying the same analysis techniques on the context distribution shift as they did on the reward distribution?

**Limitations:**

The authors discuss limitations of their work in a separate Discussion section. An important one that the authors mention is how to choose the maximum divergence, which would affect how conservative the optimized policy will be.

**Strengths And Weaknesses:**

Strengths:
- The paper tackles an important problem of robust batch policy optimization. The authors provide a compelling motivation for their work via voting and healthcare applications.
- The proposed algorithm is sound and provides substantive improvements over existing baselines, including one that considers joint distribution shift rather than the context and reward shifts separately. The authors also provide an interesting approach to extend their approach beyond tabular environments via estimating the moments of the distribution.
- The empirical evaluation is comprehensive. The authors compare their proposed approach against strong, well-tuned baselines in multiple domains that rely on real-world classification and voting data.

Weaknesses:
- The exact algorithm somewhat lacks in novelty, as it is a hierarchical application of the dual objective trick used in prior works, notably Si et al. (2020). This also applies to the analysis, as Lemma 1 matches Theorem 1 of Si et al. (2020).
- To my knowledge, the theoretical guarantees (Theorem 2 and 3)  argue mostly about asymptotic convergence. The results would be stronger if there were sample complexity bounds and showed the dependence on the number of samples n. There also is a lack of guarantees on lower-bounds.
- Though the experiments use real-world data, the actual distribution shifts are simulated. It would be interesting to use a real-world example of a distribution shift.

---

> ### Author Response · Authors · 2022-08-02
> **Response to Reviewer kRMP**
>
> Thank you for your helpful comments!
> - **Real-world shifts:** Thank you for the nice suggestion! We updated Figure 1c for the Voting dataset to reflect a real-world shift that occurs between cities in the dataset. We find similar results as before where FDRO's performance increases on the test distribution as we increase the robustness parameter.
>
> - **Theoretical Analysis:** Thank you for highlighting the need for updated theoretical analysis. We updated our theoretical analysis and we have updated Theorem 2 and 3 to include rates of convergence that depend on the number of samples n. Our analysis is non-asymptotic and we show our algorithm has $\sqrt{\frac{log(n)}{n}}$ dependence on n. We no longer utilize Lemma 1 (which was Theorem 1 of Si et. al (2020)) and our updated analysis uses different techniques. For example, Theorem 1 of Si et. al uses asymptotic techniques such as the central limit theorem to give asymptotic rates of convergence for policy evaluation while our updated analysis is non-asymptotic. We agree lower bounds would be a great avenue for future work.
>
> - **Choosing the maximum divergence:** Thank you for bringing up this point. We agree this is an important issue and indeed it is a limitation common to all distributionally robust methods. To address this further, we update Figure 1f and the Appendix (Figure 3a) to include plots showing the performance of the Scene and simulated Bernoulli environments for different values of the reward shift radius hyperparameter, $\delta_c$ (0.001, 0.1, 0.25). Empirically we found performance to not vary too much based on the choice of $\delta_c$.
>
> **Did the above answer your questions? We also welcome additional questions.**

---

> > ### Comment · Reviewer_kRMP · 2022-08-08
> > **Thank you for the response.**
> >
> > The response addressed my primary concerns, and filled all the technical "holes" I could point out in the review. I do think much of the technical content overlaps quite a bit with the work by Si et al. (2020), but the authors have demonstrated that there are important benefits to using the factorized distribution. Based on its contributions, I believe the paper should be accepted, and will maintain my positive score.

---

### Author Response · Authors · 2022-08-02
**Summary of Changes in Rebuttal Version**

We thank all the reviewers for their helpful feedback! We highlight here some general changes in the rebuttal version of our submission:

- **Updated Theoretical Analysis**: We updated our theoretical analysis and we have updated Theorem 2 and 3 to include rates of convergence which depend on the number of samples n. Our analysis is non-asymptotic and we show our algorithm has $\sqrt{\frac{log(n)}{n}}$ dependence on n.

- **Updated Plots**: We updated Figure 1c for the Voting dataset to reflect a real-world shift that occurs between cities in the dataset. We find similar results as before where FDRO's performance increases on the test distribution as we increase the robustness parameter. We additionally update Figure 1f and the Appendix (Figure 3a) to include plots showing the performance in the Scene and simulated Bernoulli environments for different values of the reward shift radius hyperparameter, $\delta_c$ (0.001, 0.1, 0.25). Empirically we found performance to not vary too much based on the choice of $\delta_c$.

- **Updated Discussions**: We update our discussions in Section 6.2 to provide more discussion comparing baseline-DRO and FDRO as well as the equivalence of solutions (under the heading 'Discussion: When to use FDRO vs baseline-DRO (equivalence of solutions)).

---

### Meta-Review · Area_Chair_b1gL · 2022-08-25

**Recommendation:** Accept
**Confidence:** Certain

**Metareview:**

This paper studies off-policy learning with an environment shift, where the distributions of both contexts and rewards can change. The authors address both challenges in a factored form, and derive error bounds for both off-policy evaluation and optimization. The proposed approach is evaluated on real-world datasets. The original ratings of the paper were 6, 6, 6, 6, and 5; and they did not change after the rebuttal. The reviewers generally praised the paper and their main concerns were addressed in the rebuttal:

* Asymptotic errors bounds were replaced with finite-sample guarantees.

* Synthetic shifts in the distributions in experiments were complemented with actual shifts in data.

This is a good paper to accept and I believe that the authors will improve the paper further based on the feedback of the reviewers.

**Award:**

No

---

### Decision · Program_Chairs · 2022-09-14

Accept